# MetaKoopman: Bayesian Meta-Learning of Koopman Operators for Modeling Structured Dynamics under Distribution Shifts

**Mahmoud Selim**[1,2]    **Sriharsha Bhat**[2]    **Karl H. Johansson**[2]

[1]TRATON, `mahmoud.selim@se.traton.com`

[2]KTH Royal Institute of Technology, {`mase2, svbhat, kallej`}`@kth.se`

## Abstract

Modeling and forecasting nonlinear dynamics under distribution shifts is essential for robust decision-making in real-world systems. In this work, we propose MetaKoopman, a Bayesian meta-learning framework for modeling nonlinear dynamics through linear latent representations. MetaKoopman learns a Matrix Normal-Inverse Wishart (MNIW) prior over the Koopman operator, enabling closed-form Bayesian updates conditioned on recent trajectory segments. Moreover, it provides a closed-form posterior predictive distribution over future state trajectories, capturing both epistemic and aleatoric uncertainty in the learned dynamics. We evaluate MetaKoopman on a full-scale autonomous truck and trailer system across a wide range of adverse winter scenarios—including snow, ice, and mixed-friction conditions—as well as in simulated control tasks with diverse distribution shifts. MetaKoopman consistently outperforms prior approaches in multi-step prediction accuracy, uncertainty calibration and robustness to distributional shifts. Field experiments further demonstrate its effectiveness in dynamically feasible motion planning, particularly during evasive maneuvers and operation at the limits of traction. Project website: https://mahmoud-selim.github.io/MetaKoopman/

## 1 Introduction

Modern autonomous systems must operate in complex, nonstationary environments, where the underlying system dynamics are neither fixed nor fully known. Terrain properties can shift from asphalt to ice; payload distributions may change with each mission; mechanical degradation or actuator latency can accumulate over time. Learning a single predictive model that accounts for all such variability is difficult—if not infeasible. These sources of variation induce what is known as distributional shift: a mismatch between the training conditions used to learn a model and the test-time conditions in which it is deployed. In safety-critical applications such as autonomous driving or robotic control, even small prediction errors can compound, leading to unstable or unsafe behavior. These risks are further amplified when models are unable to assess or convey uncertainty in their predictions. This raises a fundamental question: how can we design learning-based systems that remain both accurate and reliable when the dynamics themselves evolve?

From a modeling perspective, this challenge exposes a central limitation of standard approaches: the assumption of a globally valid dynamics model that remains accurate across diverse environments and latent conditions. In practice, such models are exceptionally difficult—if not impossible—to obtain. Real-world dynamics are inherently time-varying due to hardware wear, variable payloads, and environmental factors that are either unobserved or partially observable at best. Meta-learning [2] provides one path forward by training models to quickly adapt to new conditions using limited data. However, many meta-learning frameworks rely on gradient-based inner loops or iterative optimization procedures [10, 11, 45], which can introduce latency and computational overhead that make them

39th Conference on Neural Information Processing Systems (NeurIPS 2025).

unsuitable for real-time deployment. In contrast, Koopman operator theory [16] represents nonlinear dynamics through linear latent-space operators, enabling highly efficient multi-step prediction and seamless integration with classical control and planning methods. Yet Koopman-based models are typically trained offline and remain static at test time [43, 33, 3, 1, 17], limiting their ability to respond to changing dynamics. This motivates our approach: a model that integrates the structured efficiency of Koopman operators with the adaptivity and data efficiency of meta-learning, while enabling online updates and principled uncertainty quantification.

In this work, we introduce **MetaKoopman**, a Bayesian meta-learning framework for modeling dynamics under distribution shift. At its core, MetaKoopman places a Matrix Normal–Inverse Wishart (MNIW) prior over the Koopman operator, which is meta-learned across diverse environments. In doing so, MetaKoopman learns a distribution over the underlying dynamics, capturing uncertainty in system evolution. During deployment, this prior is adapted online through closed-form Bayesian updates using recent trajectory data to obtain an updated belief (the posterior) over the dynamics. This posterior is then used to generate probabilistic forecasts (posterior predictive distribution) over future states, allowing the model to quantify both epistemic and aleatoric uncertainty in its predictions. The resulting uncertainty-aware model is integrated into a sampling-based motion planner via a variational action encoder, enabling the efficient generation of dynamically feasible trajectories. MetaKoopman is evaluated across a suite of simulated benchmarks that expose models to varying terrain slopes, friction conditions, and payload dynamics. In all cases, it achieves superior multi-step prediction accuracy, calibrated uncertainty, and robustness to distributional shifts. Beyond simulation, we deploy MetaKoopman on a 37.5-ton autonomous truck–trailer system operating in harsh winter conditions, including snow, ice, and mixed-friction surfaces. In one safety-critical test, where a virtual obstacle appears on an icy road, non-adaptive models mispredict braking feasibility and fail to react safely. In contrast, MetaKoopman rapidly infers changing surface dynamics and initiates a lane change, demonstrating how structured Bayesian adaptation enables safer and more reliable control in real-world autonomous systems.

Our contributions can be summarized as follows:

1. **Bayesian meta-learning of Koopman operators –** First to combine Koopman structure with Bayesian meta-learning, giving analytic test-time posterior over the Koopman operator. In addition, we derive a closed-form posterior-predictive distribution, enabling principled uncertainty quantification in forecasts.

2. **Meta-learned tempering for adaptation –** A scalar $\beta$, meta-learned during meta-training, rescales the MNIW prior precision at test time. This simple yet effective mechanism increases responsiveness to online adaptations while maintaining calibrated uncertainty.

3. **Variational action encoder –** Latent-action sampling accelerates trajectory rollouts by an order of magnitude, removing the main model-based planning bottleneck.

4. **Comprehensive evaluation –** We collect a truck dynamics dataset in adverse weather (snow, ice, and mixed friction) and use it with five simulated environments exhibiting distribution shifts to demonstrate state-of-the-art accuracy, calibrated uncertainty, and robustness.

5. **Real-world deployment –** Live 37.5-ton truck trials show adaptive braking on ice and safe evasive snow maneuvers where static models fail.

## 2 Related Work

**Koopman operator theory** [16] provides a linear, though infinite-dimensional, framework for analyzing nonlinear dynamical systems by lifting them into a space of observable functions, where their evolution can be described linearly. Early methods such as dynamic mode decomposition (DMD) [31, 32] and extended dynamic mode decomposition (EDMD) [42, 19] approximate the operator using manually selected observables, a process that is often brittle and task-specific. To address this, recent advances leverage deep learning to automatically learn expressive embeddings [20, 26, 43, 37], leading to neural extensions such as deep-DMD and deep-Koopman, which significantly enhance the modeling capacity and scalability of Koopman-based methods. These developments have enabled accurate prediction and control across a range of domains, particularly where linearity allows the use of classical controllers such as LQR and MPC [27, 21, 17, 1]. Applications span soft robotics [3, 33], aerial robotics [34, 22], robotic manipulation [4], and vehicle dynamics [6, 40], demonstrating the versatility of Koopman-based modeling and control across diverse nonlinear systems.

Recent studies have further investigated adaptive and time-varying Koopman operators for modeling systems with evolving dynamics. For example, an adaptive Koopman embedding was introduced to augment a nominal Koopman model with an online neural module that updates operator matrices in real time, improving robustness to parametric variations and external disturbances in control tasks [35]. Similarly, the Koopman Neural Forecaster (KNF) [41] employs global and local operators to capture both stable and evolving dynamics in non-stationary systems. However, their approach is confined to time-series forecasting and does not account for control inputs. Other approaches explore explicitly time-varying Koopman operators for modeling changing systems [46, 14], but these typically require online optimization and lack principled uncertainty estimation. In contrast, our method enables closed-form Bayesian adaptation using a meta-learned prior, providing both computational efficiency and principled uncertainty quantification.

**Meta-learning** provides a framework for learning models that rapidly adapt to new tasks by leveraging experience across related tasks [10, 9, 25, 29]. In dynamical systems, meta-learning has been applied to learn models that adapt online to changes in dynamics or environment [24, 30, 7, 8]. Approaches such as Bayesian MAML [45], PLATIPUS [11], and PEARL [28] demonstrate that learning a prior over model parameters enables fast adaptation from limited recent experience, addressing the challenge of distributional shift in real-world domains. Critically, these methods avoid the need for globally accurate dynamics models by focusing on fast, local adaptation—enabling robustness to perturbations such as hardware failures, terrain variation, or sensor noise. However, many approaches either rely on optimization-based adaptation, which is computationally expensive at test time, or lack principled uncertainty quantification. In contrast, our method performs closed-form Bayesian adaptation from short trajectory segments, combining computational efficiency with tractable uncertainty estimation.

## 3 Preliminaries

### 3.1 Modeling Nonlinear Dynamical Systems with the Koopman Operator

We consider a discrete-time, nonlinear dynamical system governed by the state evolution

$$x_{t+1} = f(x_t, u_t), \tag{1}$$

where $x_t \in \mathbb{R}^n$ denotes the system state, $u_t \in \mathbb{R}^m$ denotes the control input, and $f$ is an unknown nonlinear function. The objective is to predict future states $x_{t+1}, x_{t+2}, \ldots, x_{t+h}$ over a horizon $h$, given the current state and current and future control inputs.

The Koopman operator, denoted as $\bar{\mathcal{K}} : \bar{\mathcal{F}} \to \bar{\mathcal{F}}$, is an infinite-dimensional linear operator that describes the evolution of a nonlinear dynamical system. Here, $\bar{\mathcal{F}}$ refers to the set of all *measurement functions* or *observables*, which form an infinite-dimensional Hilbert space. More specifically, given an observable function $\psi$, the evolution of the system using the Koopman operator can expressed as:

$$\bar{\mathcal{K}}\psi(x_t, u_t) = \psi(f(x_t, u_t), u_{t+1}) = \psi(x_{t+1}, u_{t+1}) \tag{2}$$

In principle, the Koopman operator provides a complete linear description of the system's dynamics in the lifted space. However, operating directly in the infinite-dimensional Hilbert space $\bar{\mathcal{F}}$ is computationally intractable. In the following section, we introduce a finite-dimensional approximation of the Koopman operator that can be learned from data.

### 3.2 Meta-Learning for Fast Adaptation Across Trajectories

Meta-learning provides a framework for training models that can rapidly adapt to new tasks using limited data. In our setting, a task $\mathcal{T}$ corresponds to a short trajectory segment collected from a particular operating condition—such as a specific road surface, slope, or mass distribution—where the goal is to predict future states conditioned on recent state-action history. Let $\rho(\mathcal{T})$ denote a distribution over such tasks, each defined by a trajectory dataset $\mathcal{D}_\mathcal{T} = \{(x_t, u_t)\}$ sampled from a distinct environment.

The meta-learning objective is to find shared model parameters $\theta$ and an adaptation procedure $\mathcal{A}$ on an adaptation dataset $\mathcal{D}_\mathcal{T}^{tr}$, such that the adapted parameters $\theta' = \mathcal{A}(\mathcal{D}_\mathcal{T}^{tr}, \theta)$ minimize a supervised loss on a held-out query set $\mathcal{D}_\mathcal{T}^{test}$, drawn from the same task:

$$\min_{\theta, \mathcal{A}} \mathbb{E}_{\mathcal{T} \sim \rho(\mathcal{T})} [\mathcal{L}_\mathcal{T}(\mathcal{D}_\mathcal{T}^{test}, \theta')] \quad s.t. \quad \theta' = \mathcal{A}(\mathcal{D}_\mathcal{T}^{tr}, \theta). \tag{3}$$

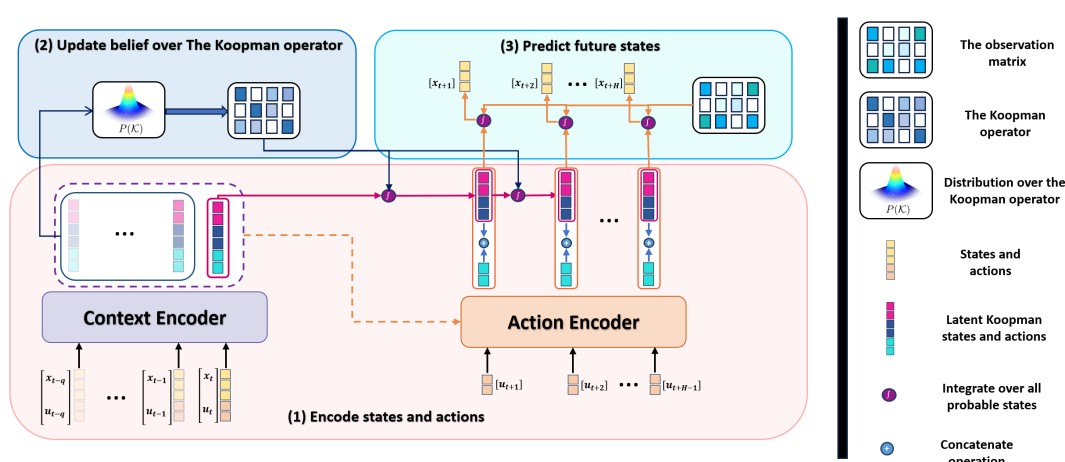

Figure 1: An Overview of the MetaKoopman framework. (1) Past states and actions are encoded into a latent representation using the context encoder. (2) This latent representation is used to update the posterior belief over the Koopman operator. (3) The updated operator is used to predict future latent states, which are decoded back to the original state space for forecasting and planning.

This formulation enables the model to generalize across environments by learning a shared initialization—or prior—that can be quickly adapted to new environments from a small number of observations. In our case, the parameters $\theta$ define a probabilistic prior over the Koopman operator, which is updated online using recent trajectory data. This structure supports task-specific, uncertainty-aware predictions, and forms the foundation of the MetaKoopman framework described next.

# 4 The MetaKoopman Framework

We now present the MetaKoopman framework, which combines Koopman-based modeling with Bayesian meta-learning to enable adaptive and uncertainty-aware dynamics prediction. The method is built around a latent linear dynamical system, where the Koopman operator is inferred via closed-form Bayesian updates from a meta-learned prior. We first describe the model structure (Section 4.1), then present Bayesian inference and predictive forecasting (Section 4.2), followed by meta-learning of the prior (Section 4.3), and finally its integration with a real-time planner (Section 4.4).

## 4.1 Latent Koopman Dynamics Model

Building on the preliminaries, directly operating with the infinite-dimensional Koopman operator is infeasible in real-world systems. To make Koopman modeling tractable, we learn a finite-dimensional approximation to the embedding function $g : \mathbb{R}^n \times \mathbb{R}^m \to \mathbb{R}^d$, where $d \gg n + m$, which lifts state-action pairs into a latent space where the dynamics evolve approximately linearly.

We denote the resulting embedding as $\tilde{z}_t = g(x_t, u_t) \in \mathbb{R}^d$, and decompose it into two parts: a state-related embedding $\tilde{x}_t \in \mathbb{R}^\eta$ and a control-related embedding $\tilde{u}_t \in \mathbb{R}^{d-\eta}$, such that $\tilde{z}_t = (\tilde{x}_t, \tilde{u}_t)$. The evolution of the system is then modeled linearly in latent space as:

$$\tilde{x}_{t+1} = \mathcal{K}\tilde{z}_t, \tag{4}$$

where $\mathcal{K} \in \mathbb{R}^{\eta \times d}$ is a finite-dimensional Koopman operator. Unlike prior methods that embed each state-action pair independently, we learn embeddings that capture temporal dependencies across multiple steps. Specifically, we define a sequence-based embedding over delayed inputs:

$$[\tilde{z}_{t-q}, \ldots, \tilde{z}_t] = G_\theta(x_{t-q}, u_{t-q}, \ldots, x_t, u_t), \tag{5}$$

We refer to $G_\theta$ as the *context encoder*, as it maps a window of past states and actions into a compact embedding that summarizes the local dynamics. This sequence-based representation captures temporal dependencies that help the Koopman operator more accurately linearize the system evolution.

These temporally enriched embeddings provide a more informative representation of local dynamics than single-step inputs, supporting more accurate modeling and inference. The use of delayed coordinates is further supported by Takens's theorem [38], which motivates the use of time-delay embeddings for reconstructing system dynamics. We analyze the effect of history length in the appendix, and find that moderate windows provide the best tradeoff between adaptability and predictive accuracy.

To perform multi-step prediction, we additionally encode future control inputs $u_{t+1}, \ldots, u_{t+h}$ using a dedicated *action encoder*. This ensures that the action embeddings are consistent with the latent dynamics inferred from the recent trajectory context. Finally, the predicted latent states are mapped back to the original state space using a learned linear observation matrix $\mathcal{C} \in \mathbb{R}^{n \times \eta}$:

$$\hat{x}_t = \mathcal{C}\tilde{x}_t. \tag{6}$$

An overview of the high-level model architecture is provided in Figure 1; additional implementation details are deferred to the Appendix.

## 4.2 Closed-Form Bayesian Inference for Koopman Dynamics

To support data-efficient and uncertainty-aware adaptation, we formulate Koopman operator learning as a Bayesian inference problem. Rather than estimating a fixed operator, we place a distribution over plausible operators and update it using recent trajectory segments. This enables principled uncertainty quantification in both the model's parameters and its predictions, and facilitates rapid adaptation to new environmental conditions.

**Probabilistic Formulation.** We extend the classical Koopman model to account for process noise, measurement error, and embedding imperfections. Specifically, we model latent dynamics as:

$$\tilde{x}_{t+1} = \mathcal{K}\tilde{z}_t + \epsilon, \quad \epsilon \sim \mathcal{N}(0, \Sigma), \tag{7}$$

Where $\mathcal{K} \in \mathbb{R}^{\eta \times d}$ is the stochastic Koopman operator, and $\Sigma \in \mathbb{R}^{\eta \times \eta}$ is the process noise covariance. Under the assumption of i.i.d. noise, the likelihood of latent transitions $\tilde{z}_t \to \tilde{x}_{t+1}$ is:

$$p(\tilde{X} \mid \mathcal{K}, \tilde{Z}, \Sigma) = \prod_{i=1}^{N} \mathcal{N}(\tilde{x}_{t+1}^i \mid \mathcal{K}\tilde{z}_t^i, \Sigma), \tag{8}$$

where $\tilde{X} \in \mathbb{R}^{\eta \times N}$, $\tilde{Z} \in \mathbb{R}^{d \times N}$, and $N$ is the number of data points. The conjugate prior for this likelihood is the matrix normal inverse Wishart distribution $\mathcal{MNIW}$:

$$\mathcal{K}, \Sigma \sim \mathcal{MNIW}(\breve{M}, \breve{V}, \breve{\nu}, \breve{\Psi}), \tag{9}$$

where $\mathcal{K}|\Sigma \sim \mathcal{MN}(\breve{M}, \Sigma, \breve{V})$, and $\Sigma \sim \mathcal{IW}(\breve{\nu}, \breve{\Psi})$.

**Lemma 4.1.** *Given the likelihood (Eq.8), the posterior over $\mathcal{K}, \Sigma$ remains in the $\mathcal{MNIW}$ family:*

$$\mathcal{K}, \Sigma \sim \mathcal{MNIW}(\hat{M}, \hat{V}, \hat{\nu}, \hat{\Psi}), \tag{10}$$

*with the posterior parameters given by:*

$$\hat{M} = S_{xz}S_{zz}^{-1}, \qquad \hat{V} = S_{zz}, \qquad \hat{\nu} = N + \breve{\nu}, \qquad \hat{\Psi} = \breve{\Psi} + S_{x|z} \tag{11}$$

*and*

$$S_{xz} = \tilde{X}\tilde{Z}^\top + \breve{M}\breve{V}, \quad S_{zz} = \tilde{Z}\tilde{Z}^\top + \breve{V}, \quad S_{xx} = \tilde{X}\tilde{X}^\top + \breve{M}\breve{V}\breve{M}^\top,$$
$$S_{x|z} = S_{xx} - S_{xz}S_{zz}^{-1}S_{xz}^\top \tag{12}$$

*Proof.* The proof follows directly from [23]. □

**Lemma 4.2.** *Given the posterior in Eq. (10), the one-step posterior predictive distribution can be well-approximated by a multivariate Gaussian:*

$$\tilde{x}_{t+1}|\tilde{z}_t, \tilde{X}, \tilde{Z} \sim \mathcal{N}\left(\hat{M}\tilde{z}_t, \hat{\Psi}\left(1 + \tilde{z}_t^\top \hat{V}\tilde{z}_t\right)\right) \tag{13}$$

*and the multi-step predictions for future latent states can be approximated by a moment-matched Gaussian, obtained by recursively propagating the first two moments and re-approximating by a Gaussian at each step:*

$$\mu_{t+1} = \hat{M}\,\mu_{z,t}, \tag{14a}$$

$$\Sigma_{t+1} = \hat{M}\,\Sigma_{z,t}\,\hat{M}^\top + \hat{\Psi}\left(1 + \mu_{z,t}^\top \hat{V}\,\mu_{z,t} + \text{tr}(\hat{V}\,\Sigma_{z,t})\right). \tag{14b}$$

*where $(\mu_t, \Sigma_t)$ denote the latent state, $\tilde{x}_t$, mean and covariance and $(\mu_{z,t}, \Sigma_{z,t})$ denote the mean and covariance of the regressor $\tilde{z}_t$.*

*Proof.* See the appendix for the derivation and discussion of the Gaussian approximation. $\square$

### 4.3 MetaKoopman: Bayesian Meta-Learning of the Koopman Operator

The Bayesian formulation in Section 4.2 enables posterior inference over the Koopman operator. However, this relies on the quality of the prior. Rather than specifying it manually, MetaKoopman *meta-learns* the prior parameters across a distribution of tasks. This yields a prior that supports fast, uncertainty-aware adaptation via closed-form Bayesian updates. This section describes the meta-learning procedure (Section 4.3.1) and the test-time adaptation strategy with tempering (Section 4.3.2).

#### 4.3.1 Meta-Learning the Prior

We parameterize the prior over the Koopman operator and noise covariance as a $\mathcal{MNIW}$ distribution with parameters $\theta = \{\check{M}, \check{V}, \check{\nu}, \check{\Psi}\}$. The goal is to meta-learn them across a distribution of tasks $\mathcal{T} \sim \rho(\mathcal{T})$, where each task corresponds to a trajectory segment from a distinct operating condition.

For each task $\mathcal{T}$, we split its trajectory data into an adaptation set $\mathcal{D}_{\mathcal{T}}^{\text{tr}}$ and a held-out evaluation set $\mathcal{D}_{\mathcal{T}}^{\text{test}}$. Using Eq. 10, we compute posterior parameters from $\mathcal{D}_{\mathcal{T}}^{\text{tr}}$, then evaluate the posterior predictive (Eq. 14) on $\mathcal{D}_{\mathcal{T}}^{\text{test}}$. The meta-objective is minimize the negative log likelihood of the ground truth under the posterior predictive distribution:

$$\min_{\theta} \; \mathbb{E}_{\mathcal{T}\sim\rho(\mathcal{T})}\left[\mathcal{L}_{\mathcal{T}}^{\text{test}}\left(\theta'(\mathcal{D}_{\mathcal{T}}^{\text{tr}})\right)\right] = \min_{\theta} \; \mathbb{E}_{\mathcal{T}\sim\rho(\mathcal{T})}\left[-\log p\left(\mathcal{D}_{\mathcal{T}}^{\text{test}} \mid \theta'(\mathcal{D}_{\mathcal{T}}^{\text{tr}})\right)\right], \tag{15}$$

where $\theta'(\mathcal{D}_{\mathcal{T}}^{\text{tr}})$ denotes the posterior parameters obtained by updating the prior $\theta$ with $\mathcal{D}_{\mathcal{T}}^{\text{tr}}$. This formulation encourages learning a prior that is both expressive and adaptable: it captures structure shared across tasks, while allowing efficient posterior inference from limited data. Unlike gradient-based meta-learning methods, adaptation here is fully analytical, requiring no inner-loop optimization.

#### 4.3.2 Online Adaptation with Tempering

To modulate the strength of prior information during adaptation, we introduce a scalar tempering factor $\beta \in (0, 1]$ that scales the prior precision. Specifically, we temper the prior as:

$$\check{V}' = \beta^{-1}\check{V}, \quad \check{\Psi}' = \beta^{-1}\check{\Psi}. \tag{16}$$

Smaller values of $\beta$ reduce the influence of the prior, allowing the model to adapt more aggressively to new data. Importantly, we *meta-learn* $\beta$ jointly with the MNIW prior parameters, treating it as a learnable scalar optimized via the meta-objective in Eq. 15. This allows MetaKoopman to discover an optimal tradeoff between prior knowledge and new evidence, tailored to the task distribution. Algorithm 1 details the procedure for meta-learning the Koopman prior.

### 4.4 Uncertainty-Aware Planning with Variational Action Encoding

To integrate MetaKoopman into a sampling-based motion planner, we must generate a large number of dynamically feasible trajectories in real time. This typically requires sampling actions and encoding them into latent space using the *action encoder*, but doing so for each sample is computationally expensive, which can bottleneck the planning process.

To mitigate this, we introduce a *variational action encoder* that enables efficient sampling of future action embeddings directly from a standard Gaussian distribution. Instead of deterministically

**Algorithm 1** Meta-Learning of the Koopman Prior

---

**Require:** Distribution over tasks $\rho(\mathcal{T})$; initial prior parameters $\theta = \{\breve{M}, \breve{V}, \breve{\nu}, \breve{\Psi}, \beta\}$
1: **while** not converged **do**
2:     Sample batch $\{\mathcal{T}_i\}_{i=1}^B \sim \rho(\mathcal{T})$
3:     **for** each task $\mathcal{T}_i$ **do**
4:         Split data into $\mathcal{D}_{\mathcal{T}_i}^{\text{tr}}, \mathcal{D}_{\mathcal{T}_i}^{\text{test}}$.
5:         Temper the prior: $\breve{V}' = \beta^{-1}\breve{V}, \quad \breve{\Psi}' = \beta^{-1}\breve{\Psi}$ using Eq. (16).
6:         Compute posterior parameters using Eq. (10) on $\mathcal{D}_{\mathcal{T}_i}^{\text{tr}}$.
7:         Evaluate log-likelihood on $\mathcal{D}_{\mathcal{T}_i}^{\text{test}}$ using Eq. (14).
8:     **end for**
9:     Update $\theta$ using gradients of total NLL across tasks (Eq. (15)).
10: **end while**

---

mapping each future action to a latent embedding, the encoder models a distribution $q_\phi(\tilde{u}_i \mid u_i)$ over latent actions $\tilde{u}_i$, parameterized by a mean and diagonal covariance matrix, where $\phi$ denotes the parameters of the variational action encoder. Following the standard VAE [15], We regularize this distribution during training to match a standard normal prior $p(\tilde{u}_i) = \mathcal{N}(0, I)$ by minimizing the KL divergence:

$$\mathcal{L}_{\text{total}} = \mathcal{L}_{\text{NLL}} + \sum_{i=t}^{t+H-1} D_{\text{KL}}\big(q_\phi(\tilde{u}_i \mid u_i) \,\big\|\, \mathcal{N}(0, I)\big). \tag{17}$$

This encourages the latent future action space to follow a standard Gaussian, enabling direct sampling $\tilde{u}_i \sim \mathcal{N}(0, I)$ at test time without evaluating the *action encoder*. This allows the planner to efficiently generate thousands of trajectories by simulating them only through the linear Koopman dynamics.

We refer to this mechanism as *Cost-Guided Latent Action Sampling (CLAS)*, which enables cost-based scoring of sampled trajectories while leveraging uncertainty-aware predictions. Implementation details and ablations are provided in the appendix.

## 5 Experimental Results

### 5.1 Experimental Setup

Our experiments aim to evaluate the capabilities of MetaKoopman in both simulated and real-world control settings. We investigate the following key questions: (i) How accurately does the model predict future dynamics under distribution shift? (ii) Does adaptation from short trajectory segments meaningfully alter the model's predictions? (iii) How does MetaKoopman compare to existing adaptive and probabilistic baselines? (iv) Can the model support safer closed-loop control in real-world deployment scenarios?

To explore these questions, we evaluate MetaKoopman across a suite of simulation benchmarks based on MuJoCo [39], as well as a full-scale real-world autonomous truck. Additional details on the truck dataset collection, along with ablations and detailed comparisons—including the effect of tempering, uncertainty calibration, computational complexity, the variational action encoder, and history length sensitivity—are provided in the appendix.

**Evaluation environments.** We evaluate MetaKoopman on a diverse set of simulation and real-world environments, each designed to test generalization under distribution shift or adaptation to structural perturbations:

- **Real-World Autonomous Truck in adverse winter conditions:** A 37.5-ton autonomous truck is evaluated under winter conditions including snow, ice, and mixed-friction roads, with shifts in tire-surface interaction and braking dynamics.

- **HalfCheetah-Slope:** The agent is trained on mild inclines ($[-10°, 10°]$) and tested on steeper, unseen slopes ($[-15°, 15°]$), evaluating generalization across varying terrain.

- **Hopper-Gravity:** Gravity is scaled during training ($[0.9, 1.1]$) and further perturbed at test time ($[0.85, 1.15]$), simulating changes in payload or environmental conditions.

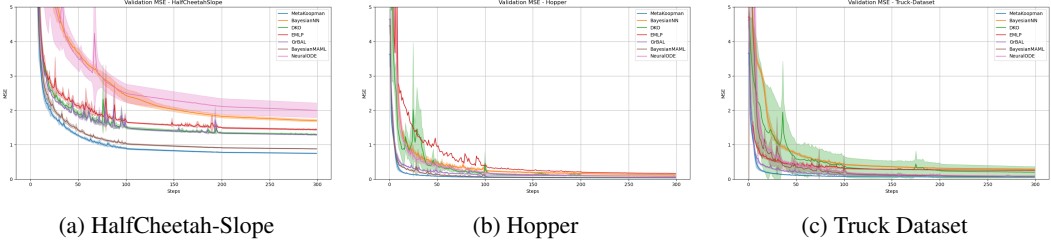

|  | (a) HalfCheetah-Slope | (b) Hopper | (c) Truck Dataset |

Figure 2: Validation MSE over time across selected environments. MetaKoopman consistently achieves lower prediction error and faster convergence compared to baselines.

Table 1: Final validation loss ($\pm$ std) across methods and environments. Best performance per environment is **bolded**.

| Environment | MetaKoopman | BayesianMAML | GrBAL | DKO | BayesianNN | EMLP | NeuralODE |
|---|---|---|---|---|---|---|---|
| Hopper | **0.0369 $\pm$ 0.0019** | 0.0475 $\pm$ 0.0027 | 0.0926 $\pm$ 0.0016 | 0.0913 $\pm$ 0.0051 | 0.1332 $\pm$ 0.0116 | 0.1683 $\pm$ 0.0000 | 0.0793 $\pm$ 0.0078 |
| Truck-Dataset | **0.0596 $\pm$ 0.0021** | 0.0888 $\pm$ 0.0100 | 0.0917 $\pm$ 0.0023 | 0.2042 $\pm$ 0.1674 | 0.2923 $\pm$ 0.0120 | 0.2589 $\pm$ 0.0053 | 0.0926 $\pm$ 0.0022 |
| HalfCheetah-Slope | **0.7490 $\pm$ 0.0117** | 0.8808 $\pm$ 0.0104 | 1.2800 $\pm$ 0.0028 | 1.2927 $\pm$ 0.0196 | 1.7008 $\pm$ 0.0374 | 1.4395 $\pm$ 0.0210 | 1.9981 $\pm$ 0.2123 |

- **Walker-Friction:** Surface friction is varied from $\mu/3$ to $3\mu$ during training, and extended to $[\mu/5, 5\mu]$ at test time, exposing models to extreme differences in contact dynamics (e.g., dry vs. slippery terrain).

- **Ant-DisabledJoints:** The agent is trained with one of the first six joints disabled and tested with arbitrary joint failures among all eight, modeling structural perturbations such as hardware damage or actuator failure.

- **Panda-Damping:** A 7-DoF *Panda Lift* manipulation task from *robosuite*, where joint damping coefficients are varied during training and extended beyond the seen range at test time, evaluating adaptation to shifts in contact and actuation dynamics typical of manipulation domains.

- **HalfCheetah-Stationary:** A standard unperturbed HalfCheetah environment is included to evaluate the robustness and the behavior in the absence of distribution shift, ensuring adaptive methods do not degrade under stationary conditions.

**Baselines.** We evaluate MetaKoopman by comparing it against a a range of strong and widely used baselines that span deterministic, probabilistic, and meta-learned dynamics models: (1) Deep Koopman Operator (DKO) [33], employs neural networks to jointly learn observable functions and the Koopman operator, embedding a single state-action pair rather than entire trajectories. However, it lacks adaptation and uncertainty estimation. (2) Gradient-Based Adaptive Learner (GrBAL) [24] is a model-based meta-learner that performs online adaptation via gradient updates, but offers no calibrated uncertainty. (3) Bayesian Model-Agnostic Meta-Learner (Bayesian MAML) [45] meta-learns a prior over dynamics models, combining adaptation with uncertainty estimation, but relies on optimization-based updates at test time. (4) Bayesian Neural Networks (BNNs) model predictive uncertainty through approximate posterior sampling, but do not adapt to changing dynamics. (5) Ensemble models (EMLP) [18] train an ensemble of five neural networks, estimating uncertainty through inter-model variance. They provide robustness but lack explicit mechanisms for adaptation. (6) Finally, Neural Ordinary Differential Equations (Neural ODEs) [5] model continuous-time dynamics but lack both uncertainty estimation and test-time adaptability. These baselines contextualize MetaKoopman's contributions in terms of structure, adaptability, and principled uncertainty.

## 5.2 Predictive Dynamics under Distribution Shift

We evaluate MetaKoopman's predictive accuracy on HalfCheetah-Slope, Hopper-Gravity, and a real-world truck dataset. As shown in Figure 2, MetaKoopman consistently achieves lower MSE across all environments. Table 1 further summarizes the final validation losses, where MetaKoopman outperforms other approaches in both simulated and real-world settings. The results demonstrate that the combination of structured Koopman modeling and Bayesian adaptation leads to accurate, data-efficient predictions that remain reliable under distributional shifts. Extended evaluations are provided in the appendix.

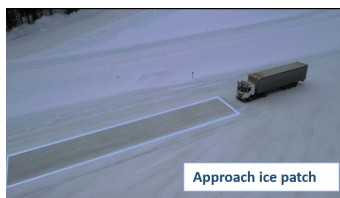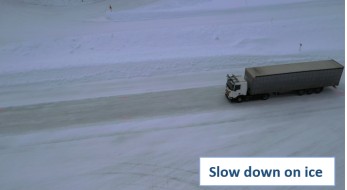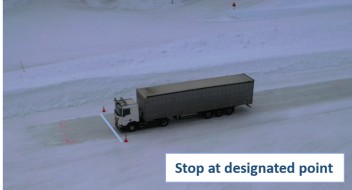

**Approach ice patch**      **Slow down on ice**      **Stop at designated point**

Figure 4: As the truck approaches a previously unobserved ice patch on a slippery road, MetaKoopman enables the planner to adapt, allowing the vehicle to brake accurately on a low friction surface.

## 5.3 Does adaptation meaningfully change the model?

To evaluate whether adaptation meaningfully alters the underlying dynamics model, we assess MetaKoopman's capability to refine its predictions after incorporating recent trajectory history. Using the real-world truck dataset, we compare the distribution of normalized mean squared error (MSE) across forecasted trajectories before and after performing a Bayesian meta-update of the Koopman operator.

As shown in Fig. 3, prior to adaptation the model exhibits higher prediction error, reflecting a mismatch between the meta-learned prior and the current operating conditions—such as changes in surface friction or tire–road interaction. After performing the Bayesian meta-update using recent trajectory data, the error distribution shifts noticeably toward lower values. This shift indicates improved alignment with the evolving system dynamics, refining its predictive model to remain consistent with the observed environment.

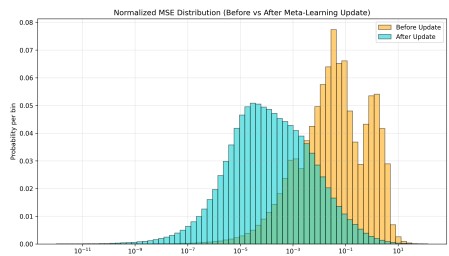

Figure 3: MSE distribution before and after Koopman operator adaptation on the truck dataset. The post-update distribution (blue) shows a shift toward lower errors compared to the pre-update (orange)

## 5.4 Closed-Loop Impact: Safer Planning Through Adaptive Uncertainty-Aware Dynamics

While learning-based dynamics models have shown promise in simulation, their application to real-world autonomous systems remains limited. A key challenge is how to evaluate not just predictive accuracy, but also how such models influence downstream decision-making—especially in conditions that involve distribution shifts, environmental uncertainty, and high-stakes planning. Do these models meaningfully adapt in time? Can they influence control choices that improve safety or robustness?

To answer these questions, we conduct a full-scale deployment on a 37.5-ton autonomous truck and trailer system during a series of tests in a northern climate during winter 2025. These tests were carried out on a closed-course proving ground designed to simulate extreme low-traction environments, including snow-covered roads, polished ice, and mixed-friction (mu-split) surfaces.[1]

**Precise stops on polished ice.** In the first scenario, the vehicle is required to stop at a designated marker after entering a zone of dramatically reduced surface friction. The road transitions from high-grip snow to polished ice, creating a distribution shift that significantly alters braking dynamics. Models that do not adapt to this shift underestimate stopping distance and consistently overshoot the target. As shown in Figure 4, MetaKoopman adapts online to the changing traction conditions, adjusting its predictive dynamics model using recent trajectory data.

**Evasive lane change on snow.** In this scenario, a virtual obstacle appears unexpectedly on a snow-covered road. The system must immediately decide whether to brake or initiate a lane change. Due to low traction, braking is physically infeasible, but non-adaptive models fail to account for this and attempt to stop, resulting in overshoot or collision. As shown in Figure 5, MetaKoopman detects the distribution shift through increased prediction error and adapts its dynamics model using recent trajectory data. This allows the planner to accurately assess braking feasibility and execute a safe

---

[1]All tests were conducted with the assistance of a professional safety driver.

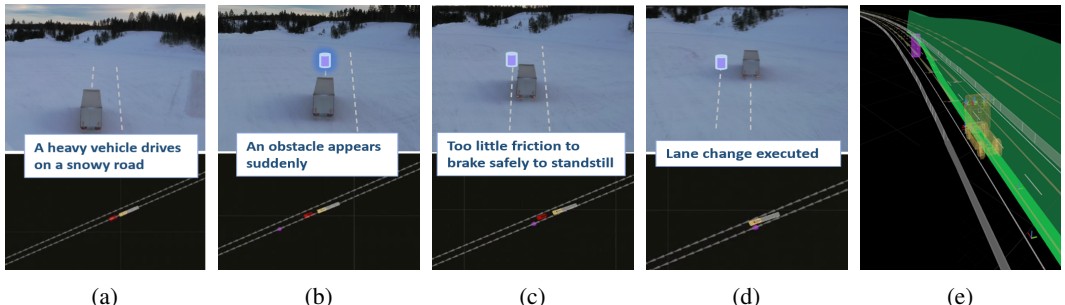

| (a) | (b) | (c) | (d) | (e) |

Figure 5: **Evasive lane change on snow. (a–d):** A virtual obstacle appears mid-lane on a low-traction road. Non-adaptive models underestimate stopping distance and attempt braking, resulting in failure. MetaKoopman adapts online and executes a safe lane change to avoid collision. **(e):** Sensor visualization showing obstacle detection and the feasible trajectory selected by the planner.

evasive maneuver. The result is a controlled lane change that avoids the obstacle while respecting vehicle dynamics and control limits. More on the real-world setup and experiments can be found in the appendix.

## 6   Discussion

MetaKoopman bridges structured modeling and meta-learning through a tractable, uncertainty-aware framework for online adaptation under distributional shift. In our formulation, uncertainty is modeled explicitly over the *Koopman operator*, while the encoder remains deterministic. This design choice is deliberate: in many practical settings—particularly those involving distributional shifts—the dominant source of variability arises from changes in system dynamics rather than in the representation itself. By keeping the encoder fixed and meta-learning only a prior over the Koopman operator, MetaKoopman focuses uncertainty modeling where it matters most—in the evolving dynamics. This assumption is justified in structured environments where the encoder is trained across diverse dynamics, yielding stable latent representations while uncertainty in the learned dynamics governs predictive quality.

Several works have explored *representation-level uncertainty* by placing distributions over latent embeddings, such as variational or stochastic encoders in dynamical systems [12, 44, 13]. Integrating such approaches could further capture uncertainty in sparse or ambiguous regions, complementing the current operator-level formulation. Overall, capturing epistemic uncertainty over the Koopman operator via the conjugate MNIW prior enables data-efficient, closed-form Bayesian updates suitable for real-time planning and control. While extending uncertainty to the representation space is a promising future direction, our results show that the present formulation already achieves strong predictive accuracy and reliable uncertainty calibration under distributional shift.

More broadly, MetaKoopman's structure suggests several directions for future work. These include integration with reinforcement learning for uncertainty-aware exploration, extension to multi-agent or partially observable environments, and application to decision-critical planning pipelines. Our results show that closed-form Bayesian meta-learning is not only tractable but viable in real-world autonomous systems, offering a practical path toward principled uncertainty in safety-critical control.

## 7   Conclusion

We introduced MetaKoopman, a Bayesian meta-learning framework for structured dynamics modeling under distribution shift. By meta-learning a conjugate prior over Koopman operators, MetaKoopman enables closed-form posterior updates from short trajectory segments and produces calibrated posterior predictive distributions for planning. Our experiments demonstrate that MetaKoopman consistently improves multi-step prediction accuracy, adapts effectively to new dynamics, and enables safer closed-loop control in real-world deployment on a heavy-duty autonomous truck. These results show that tractable Bayesian meta-learning can scale to safety-critical systems, and open the door to future extensions in uncertainty-aware control, reinforcement learning, and real-time robotics.

## Acknowledgments

This work was supported by Vinnova, Sweden, through the AllDrive project. We also want to express our gratitude to the AllDrive team for their valuable contributions, collaboration, and support throughout the winter tests.

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

# A    Posterior Predictive Distribution

Given an embedding $\tilde{z}_t$ in time $t$, the goal is to predict the probability over the state embedding for the next time-step $p(\tilde{x}_{t+1}|\tilde{z}_t, \mathcal{D})$, where $\mathcal{D}$ denotes the training dataset.

$$p(\tilde{x}_{t+1}|\tilde{z}_t, \mathcal{D}) = \iint p(\tilde{x}_{t+1}|\mathcal{K}, \Sigma, \tilde{z}_t, \mathcal{D})p(\mathcal{K}|\Sigma)p(\Sigma) \quad d\mathcal{K}d\Sigma \tag{18}$$

First, we look at $\int p(\tilde{x}_{t+1}|\mathcal{K}, \Sigma, \tilde{z}_t, \mathcal{D})p(\mathcal{K}|\Sigma)d\mathcal{K}$. We know that:

$$p(\mathcal{K}|\Sigma) = \mathcal{MN}(\hat{M}, \Sigma, \hat{V})$$
$$= \frac{1}{(2\pi)^{\frac{\eta k}{2}}|\Sigma|^{\frac{k}{2}}|\hat{V}|^{\frac{\eta}{2}}} exp\left[\frac{-1}{2}tr\left(\left(\mathcal{K} - \hat{M}\right)^{\top}\Sigma^{-1}\left(\mathcal{K} - \hat{M}\right)\hat{V}^{-1}\right)\right] \tag{19}$$

and,

$$p(\tilde{x}_{t+1}|\mathcal{K}, \Sigma, \tilde{z}_t, \mathcal{D}) = \frac{1}{(2\pi)^{\frac{\eta}{2}}|\Sigma|^{\frac{1}{2}}} exp\left[\frac{-1}{2}(\tilde{x}_{t+1} - \mathcal{K}\tilde{z}_t)^{\top}\Sigma^{-1}(\tilde{x}_{t+1} - \mathcal{K}\tilde{z}_t)\right] \tag{20}$$

Then:

$$\int p(\tilde{x}_{t+1}|\mathcal{K}, \Sigma, \tilde{z}_t, \mathcal{D})p(\mathcal{K}|\Sigma)d\mathcal{K}$$

$$= \int \frac{1}{(2\pi)^{\frac{\eta+\eta k}{2}}|\Sigma|^{\frac{1+k}{2}}|\hat{V}|^{\frac{\eta}{2}}} exp\left[\frac{-1}{2}tr\left(\Sigma^{-1}\left((\tilde{x}_{t+1} - \mathcal{K}\tilde{z}_t)(\tilde{x}_{t+1} - \mathcal{K}\tilde{z}_t)^{\top} + \left(\mathcal{K} - \hat{M}\right)\hat{V}^{-1}\left(\mathcal{K} - \hat{M}\right)^{\top}\right)\right)\right]$$

$$= \int \frac{1}{(2\pi)^{\frac{\eta+\eta k}{2}}|\Sigma|^{\frac{1+k}{2}}|\hat{V}|^{\frac{\eta}{2}}} exp\left[\frac{-1}{2}tr\left(\Sigma^{-1}\left(\mathcal{K}\left(\tilde{z}_t\tilde{z}_t^{\top} + \hat{V}^{-1}\right)\mathcal{K}^{\top} - 2\left(\tilde{x}_{t+1}\tilde{z}_t^{\top} + \hat{M}\hat{V}^{-1}\right)\mathcal{K}^{\top}\right.\right.\right.$$
$$\left.\left.\left. + \hat{M}\hat{V}^{-1}\hat{M}^{\top} + \tilde{x}_{t+1}(\tilde{x}_{t+1})^{\top}\right)\right)\right]$$

$$= \int \frac{1}{(2\pi)^{\frac{\eta+\eta k}{2}}|\Sigma|^{\frac{1+k}{2}}|\hat{V}|^{\frac{\eta}{2}}} exp\left[\frac{-1}{2}tr\left(\Sigma^{-1}\left(\mathcal{K}S_{aa}\mathcal{K}^{\top} - 2S_{ab}\mathcal{K}^{\top} + S_{bb}\right)\right)\right]$$

$$= \int \frac{1}{(2\pi)^{\frac{\eta+\eta k}{2}}|\Sigma|^{\frac{1+k}{2}}|\hat{V}|^{\frac{\eta}{2}}} exp\left[\frac{-1}{2}tr\left(\Sigma^{-1}\left(\left(\mathcal{K} - S_{ab}S_{aa}^{-1}\right)S_{aa}\left(\mathcal{K} - S_{ab}S_{aa}^{-1}\right)^{\top} + S_{a|b}\right)\right)\right]$$

$$= \frac{(2\pi)^{\frac{k\eta}{2}}|\Sigma|^{\frac{k}{2}}|S_{aa}|^{-\frac{\eta}{2}}}{(2\pi)^{\frac{\eta+\eta k}{2}}|\Sigma|^{\frac{1+k}{2}}|\hat{V}|^{\frac{\eta}{2}}} exp\left[\frac{-1}{2}tr\left(\Sigma^{-1}S_{a|b}\right)\right] = \frac{|S_{aa}|^{-\frac{\eta}{2}}}{(2\pi)^{\frac{\eta}{2}}|\Sigma|^{\frac{1}{2}}|\hat{V}|^{\frac{\eta}{2}}} exp\left[\frac{-1}{2}tr\left(\Sigma^{-1}S_{a|b}\right)\right] \tag{21}$$

where:

$$S_{aa} = \tilde{z}_t\tilde{z}_t^{\top} + \hat{V}^{-1}$$
$$S_{ab} = \tilde{x}_{t+1}\tilde{z}_t^{\top} + \hat{M}\hat{V}^{-1}$$
$$S_{bb} = \hat{M}\hat{V}^{-1}\hat{M}^{\top} + \tilde{x}_{t+1}(\tilde{x}_{t+1})^{\top}$$
$$S_{a|b} = S_{bb} - S_{ab}S_{aa}^{-1}S_{ab}^{\top}$$

Substituting Eq. 21 into Eq. 18 yields:

$$p(\tilde{x}_{t+1}|\tilde{z}_t, \mathcal{D}) = \int \frac{|S_{aa}|^{-\frac{\eta}{2}}|\hat{\Psi}|^{\frac{\hat{\nu}}{2}}}{(2\pi)^{\frac{\eta}{2}}|\Sigma|^{\frac{1}{2}}|\hat{V}|^{\frac{\eta}{2}}2^{\frac{\hat{\nu}\eta}{2}}\Gamma_\eta\left(\frac{\hat{\nu}}{2}\right)}|\Sigma|^{-\frac{\hat{\nu}+\eta+1}{2}}exp\left[\frac{-1}{2}tr\left(\Sigma^{-1}\left(S_{a|b}+\hat{\Psi}\right)\right)\right] \quad d\Sigma$$

$$= \frac{|S_{aa}|^{-\frac{\eta}{2}}|\hat{\Psi}|^{\frac{\hat{\nu}}{2}}}{(2\pi)^{\frac{\eta}{2}}|\hat{V}|^{\frac{\eta}{2}}2^{\frac{\hat{\nu}\eta}{2}}\Gamma_\eta\left(\frac{\hat{\nu}}{2}\right)}\int|\Sigma|^{-\frac{\hat{\nu}+\eta+1+1}{2}}exp\left[\frac{-1}{2}tr\left(\Sigma^{-1}\left(S_{a|b}+\hat{\Psi}\right)\right)\right] \quad d\Sigma$$

$$= \frac{|S_{aa}|^{-\frac{\eta}{2}}|\hat{\Psi}|^{\frac{\hat{\nu}}{2}}}{(2\pi)^{\frac{\eta}{2}}|\hat{V}|^{\frac{\eta}{2}}2^{\frac{\hat{\nu}\eta}{2}}\Gamma_\eta\left(\frac{\hat{\nu}}{2}\right)}\frac{2^{\frac{\hat{\nu}\eta}{2}}\Gamma_\eta\left(\frac{\hat{\nu}+\eta}{2}\right)}{|S_{a|b}+\hat{\Psi}|^{\frac{\hat{\nu}+\eta}{2}}}$$

$$= \frac{\Gamma_\eta\left(\frac{\hat{\nu}+\eta}{2}\right)|S_{aa}|^{-\frac{\eta}{2}}|\hat{\Psi}^{-\frac{\eta}{2}}|}{(2\pi)^{\frac{\eta}{2}}\Gamma_\eta\left(\frac{\hat{\nu}}{2}\right)|\hat{V}|^{\frac{\eta}{2}}}|I+\hat{\Psi}^{-1}S_{a|b}|^{-\frac{\hat{\nu}+\eta}{2}} \tag{22}$$

Now, let's break $S_{a|b}$ first. We know that:

$$S_{a|b} = S_{bb} - S_{ab}S_{aa}^{-1}S_{ab}^\top$$

$$= \hat{M}\hat{V}^{-1}\hat{M}^\top + \tilde{x}_{t+1}(\tilde{x}_{t+1})^\top - \left(\tilde{x}_{t+1}\tilde{z}_t^\top + \hat{M}\hat{V}^{-1}\right)\underbrace{\left(\tilde{z}_t\tilde{z}_t^\top + \hat{V}^{-1}\right)^{-1}}_{C}\left(\tilde{x}_{t+1}\tilde{z}_t^\top + \hat{M}\hat{V}^{-1}\right)^\top$$

$$= \hat{M}\hat{V}^{-1}\hat{M}^\top + \tilde{x}_{t+1}(\tilde{x}_{t+1})^\top - \tilde{x}_{t+1}\tilde{z}_t^\top C\tilde{z}_t(\tilde{x}_{t+1})^\top + \tilde{x}_{t+1}\tilde{z}_t^\top C\hat{V}^{-1}\hat{M}^\top + \hat{M}\hat{V}^{-1}C\tilde{z}_t(\tilde{x}_{t+1})^\top + \hat{M}\hat{V}^{-1}C\hat{V}^{-1}\hat{M}^\top$$

$$= \tilde{x}_{t+1}\underbrace{\left(I-\tilde{z}_t^\top C\tilde{z}_t\right)}_{S_{ii}}(\tilde{x}_{t+1})^\top - 2\underbrace{\hat{M}\hat{V}^{-1}C\tilde{z}_t}_{S_{ij}}(\tilde{x}_{t+1})^\top + \underbrace{\hat{M}\hat{V}^{-1}\hat{M}^\top + \hat{M}\hat{V}^{-1}C\hat{V}^{-1}\hat{M}^\top}_{S_{jj}}$$

$$= \left(\tilde{x}_{t+1}-S_{ij}S_{ii}^{-1}\right)^\top S_{ii}\left(\tilde{x}_{t+1}-S_{ij}S_{ii}^{-1}\right) + \underbrace{S_{j|i}}_{=0} \tag{23}$$

Now, we have reached a quadratic formula. We begin by using the Woodbury formula on $S_{ii}^{-1}$:

$$S_{ii}^{-1} = \left(I-\tilde{z}_t^\top C\tilde{z}_t\right)^{-1}$$

$$= I + \tilde{z}_t^\top\left(C^{-1}-\tilde{z}_t\tilde{z}_t^\top\right)^{-1}\tilde{z}_t$$

$$= I + \tilde{z}_t^\top\left(\tilde{z}_t\tilde{z}_t^\top + \hat{V}^{-1} - \tilde{z}_t\tilde{z}_t^\top\right)^{-1}\tilde{z}_t$$

$$= 1 + \tilde{z}_t^\top\hat{V}\tilde{z}_t \tag{24}$$

Then,

$$S_{ij}S_{ii}^{-1} = \hat{M}\hat{V}^{-1}C\tilde{z}_t\left(1+\tilde{z}_t^\top\hat{V}\tilde{z}_t\right)$$

$$= \hat{M}\hat{V}^{-1}\left(\hat{V}-\hat{V}\tilde{z}_t\left(1+\tilde{z}_t^\top\hat{V}\tilde{z}_t\right)^{-1}\tilde{z}_t^\top\hat{V}\right)\tilde{z}_t\left(1+\tilde{z}_t^\top\hat{V}\tilde{z}_t\right)$$

$$= \hat{M}\left(I-\tilde{z}_t\left(1+\tilde{z}_t^\top\hat{V}\tilde{z}_t\right)^{-1}\tilde{z}_t^\top\hat{V}\right)\tilde{z}_t\left(1+\tilde{z}_t^\top\hat{V}\tilde{z}_t\right)$$

$$= \hat{M}\left(\tilde{z}_t\left(1+\tilde{z}_t^\top\hat{V}\tilde{z}_t\right)-\tilde{z}_t\left(1+\tilde{z}_t^\top\hat{V}\tilde{z}_t\right)^{-1}\tilde{z}_t^\top\hat{V}\tilde{z}_t\left(1+\tilde{z}_t^\top\hat{V}\tilde{z}_t\right)\right)$$

$$= \hat{M}\left(\tilde{z}_t\left(1+\tilde{z}_t^\top\hat{V}\tilde{z}_t\right)-\tilde{z}_t\tilde{z}_t^\top\hat{V}\tilde{z}_t\left(1+\tilde{z}_t^\top\hat{V}\tilde{z}_t\right)^{-1}\left(1+\tilde{z}_t^\top\hat{V}\tilde{z}_t\right)\right)$$

$$= \hat{M}\tilde{z}_t \tag{25}$$

Substituting 25 and 23 into 22 yields:

$$p(\tilde{x}_{t+1}|\tilde{z}_t, \mathcal{D}) = \frac{\Gamma_\eta\left(\frac{\hat{\nu}+\eta}{2}\right)|S_{aa}|^{-\frac{\eta}{2}}|\hat{\Psi}|^{-\frac{\eta}{2}}}{(2\pi)^{\frac{\eta}{2}}\Gamma_\eta\left(\frac{\hat{\nu}}{2}\right)|\hat{V}|^{\frac{\eta}{2}}}|I + \hat{\Psi}^{-1}\left(\tilde{x}_{t+1} - \hat{M}\tilde{z}_t\right)^\top\left(1 + \tilde{z}_t^\top\hat{V}\tilde{z}_t\right)^{-1}\left(\tilde{x}_{t+1} - \hat{M}\tilde{z}_t\right)|^{-\frac{\hat{\nu}+\eta}{2}}$$

Using the matrix determinant lemma, this equals to:

$$p(\tilde{x}_{t+1}|\tilde{z}_t, \mathcal{D}) = \frac{\Gamma_\eta\left(\frac{\hat{\nu}+\eta}{2}\right)}{(2\pi)^{\frac{\eta}{2}}\Gamma_\eta\left(\frac{\hat{\nu}}{2}\right)}\left|\frac{\hat{\Psi}}{1 + \tilde{z}_t^\top\hat{V}\tilde{z}_t}\right|^{-\frac{\eta}{2}}\left|I + \left(\tilde{x}_{t+1} - \hat{M}\tilde{z}_t\right)^\top\frac{\hat{\Psi}^{-1}}{1 + \tilde{z}_t^\top\hat{V}\tilde{z}_t}\left(\tilde{x}_{t+1} - \hat{M}\tilde{z}_t\right)\right|^{-\frac{\hat{\nu}+\eta}{2}}$$

$$= \mathcal{T}_{\hat{\nu}}\left(\hat{M}\tilde{z}_t, \Psi\left(1 + \tilde{z}_t^\top\hat{V}\tilde{z}_t\right)\right) \tag{26}$$

Which is a multivariate Student t-distribution with mean $\hat{M}\tilde{z}_t$ and scale matrix $\Psi\left(1 + \tilde{z}_t^\top\hat{V}\tilde{z}_t\right)$.

Under large number of degrees of freedom (large number of training examples in our case), this distributions converges to a multivariate gaussian distribution with mean $\hat{M}\tilde{z}_t$ and a covariance $\Psi\left(1 + \tilde{z}_t^\top\hat{V}\tilde{z}_t\right)$.

**On the Gaussian Approximation.** The posterior predictive distribution in Eq. 26 is a multivariate Student-$t$ with degrees of freedom $\nu' = \nu_0 + N$, where $\nu_0$ is the prior's degrees of freedom and $N$ is the number of test-time adaptation points. It is well known that the Student-$t$ closely approximates a Gaussian when $\nu' > 30$. In our framework, this condition is reliably satisfied: $\nu_0$ is meta-learned and chosen to exceed the latent dimension $\eta$, which is typically at least 32, and $N$ adds further support. As a result, $\nu'$ is well above the threshold in all cases we consider, justifying the Gaussian approximation used in predictive forecasting.

**Multi-step predictive via moment matching.** We now extend the single-step posterior predictive in Eq. (26) to a multi-step forecasting rule. As shown in Eq. (26), the *conditional* one-step predictive is a multivariate Student-$t$ with mean $\hat{M}\tilde{z}_t$ and state-dependent scale $\hat{\Psi}(1 + \tilde{z}_t^\top\hat{V}\tilde{z}_t)$. For large degrees of freedom $\hat{\nu}$, we adopt the standard Gaussian surrogate

$$\tilde{x}_{t+1} \mid \tilde{z}_t \approx \mathcal{N}\left(\hat{M}\tilde{z}_t, \ \hat{\Psi}\left(1 + \tilde{z}_t^\top\hat{V}\tilde{z}_t\right)\right), \tag{27}$$

which preserves the exact first two moments of the Student-$t$. Although this marginal is generally non-Gaussian, its mean and covariance admit closed forms. We therefore follow a principled *moment matching* strategy: at each horizon step we compute the exact mean and covariance of the marginal induced by (27), and we re-approximate that marginal by the Gaussian with those moments.

**Regressor and state moments.** Let $\mu_{z,t} := \mathbb{E}[\tilde{z}_t]$, $\Sigma_{z,t} := \text{Cov}(\tilde{z}_t)$ denote the regressor moments at time $t$, and $\mu_t := \mathbb{E}[\tilde{x}_t]$, $\Sigma_t := \text{Cov}(\tilde{x}_t)$ the state moments. We use the quadratic-form identity

$$\mathbb{E}\left[\tilde{z}_t^\top\hat{V}\tilde{z}_t\right] = \mu_{z,t}^\top\hat{V}\mu_{z,t} + \text{tr}\left(\hat{V}\Sigma_{z,t}\right). \tag{28}$$

**Recursive multi-step moment matching.** To forecast $h$ steps ahead, we iterate the one-step calculation while *recomputing* the regressor moments from the current state moments at each step. Under the Gaussian surrogate (27), the *marginal* recursive moment-matched mean and covariance can be given as follows:

$$\mu_{t+1} = \mathbb{E}[\tilde{x}_{t+1}] = \hat{M}\mu_{z,t}, \tag{29}$$

$$\Sigma_{t+1} = \text{Cov}(\tilde{x}_{t+1}) = \hat{M}\Sigma_{z,t}\hat{M}^\top + \hat{\Psi}\left(1 + \mu_{z,t}^\top\hat{V}\mu_{z,t} + \text{tr}(\hat{V}\Sigma_{z,t})\right). \tag{30}$$

*Proof.* By the tower property, $\mu_{t+1} = \mathbb{E}\left[\mathbb{E}(\tilde{x}_{t+1} \mid \tilde{z}_t)\right] = \mathbb{E}[\hat{M}\mu_{z,t}] = \hat{M}\mu_{z,t}$.

For the covariance, apply the law of total variance:

$$\Sigma_{t+1} = \text{Cov}\left(\mathbb{E}[\tilde{x}_{t+1} \mid \tilde{z}_t]\right) + \mathbb{E}\left[\text{Cov}(\tilde{x}_{t+1} \mid \tilde{z}_t)\right] = \hat{M}\Sigma_{z,t}\hat{M}^\top + \hat{\Psi}\,\mathbb{E}\left[1 + \tilde{z}_t^\top\hat{V}\tilde{z}_t\right],$$

using (28), this becomes:

$$\Sigma_{t+i+1} = \hat{M}\,\Sigma_{z,t+i}\,\hat{M}^\top \;+\; \hat{\Psi}\left(1 + \mu_{z,t+i}^\top \hat{V}\mu_{z,t+i} + \text{tr}(\hat{V}\Sigma_{z,t+i})\right),$$

$\square$

## A.1 On Modeling and Interpreting Uncertainty

The posterior predictive distribution derived is a approximated with a multivariate gaussian distribution, with a scale matrix

$$\underbrace{\hat{M}\,\Sigma_{z,t}\,\hat{M}^\top}_{\text{propagation of state uncertainty}} \;+\; \underbrace{\hat{\Psi}}_{\text{aleatoric}} \times \underbrace{\left(1 + \mu_{z,t}^\top \hat{V}\mu_{z,t} + \text{tr}(\hat{V}\,\Sigma_{z,t})\right)}_{\text{epistemic modulation via }\hat{V}},$$

which naturally decomposes predictive uncertainty into two components. The matrix $\Psi$ represents *aleatoric uncertainty*, capturing irreducible system noise—such as sensor inaccuracies and actuation imprecision—that persists across test-time conditions. The multiplicative term $\mu_{z,t+i}^\top \hat{V}\mu_{z,t+i} +$ $\text{tr}(\hat{V}\Sigma_{z,t+i})$ reflects *epistemic uncertainty*, expressing the model's confidence in the latent Koopman dynamics based on limited recent data. This formulation ensures that predictive variance increases in less-familiar regions of the latent space, and contracts where the model has strong task-specific evidence.

In our framework, uncertainty is modeled explicitly over the Koopman operator, while the encoder remains deterministic. This modeling choice is guided by the observation that, in many practical settings—particularly those involving distribution shift—*model uncertainty tends to dominate*. When test-time dynamics deviate from training-time conditions (e.g., due to changing surface friction or payload configurations), it is primarily the uncertainty in the evolving system behavior—not the representation—that governs prediction quality. This is a reasonable assumption in structured systems where the encoder has been trained across a broad and diverse set of environments. In such settings, latent representations tend to be stable and informative, and the uncertainty in the learned dynamics is the principal source of variability affecting performance.

By capturing epistemic uncertainty over $K$ and leveraging the conjugate structure of the Matrix Normal-Inverse Wishart prior, we enable closed-form Bayesian updates that are both data-efficient and computationally lightweight. This design supports fast, adaptive inference suitable for real-time applications such as motion planning and control, where latency is critical. While incorporating representation-level uncertainty is a promising extension, our empirical results suggest that the current formulation provides strong predictive accuracy and reliable uncertainty estimates under challenging distribution shifts.

## B  Truck and Trailer Data Collection

One of the key contributions of this work is the application of motion planning techniques for autonomous truck and trailer systems. Autonomous driving datasets are typically expensive to acquire and maintain, with access often restricted to select (OEMs) and their suppliers. This study involved extensive data collection from a real autonomous truck and trailer platform to develop test datasets that accurately reflect the complex dynamics of these systems under diverse and challenging conditions, including harsh winter environments. Data was gathered over 12 months (Jan 2024 – Jan 2025) using a 37.5-ton, 17-meter-long Scania autonomous tractor-semitrailer in various driving and weather scenarios, covering all seasonal variations (see Fig. 6). Each session was supervised by a safety driver and test engineer to ensure safety and system reliability.

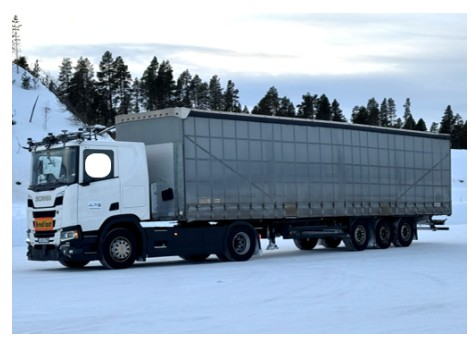

Figure 6: Truck and semi-trailer autonomous vehicle used for testing.

The dataset includes comprehensive autonomy-related logs from various high-fidelity sensors. Special emphasis was placed on capturing edge cases and distribution shifts, particularly under challenging winter conditions. Routine tests included maneuvers on a high-speed test track and test drives on public highways in different weather conditions including sun, snow and rain. Driving scenarios included straight line driving, negotiating curves and slopes, lane changes, cut-ins, stopping, following other actors and highway driving. Data collection was conducted either through fully autonomous operation or, for higher-risk maneuvers, via manual driving with onboard sensors and logging systems enabled.

To evaluate autonomous driving performance under extreme winter conditions, a series of targeted field tests were conducted during February and March 2025 on specialized proving grounds located in a northern climate. These controlled environments included a variety of challenging low-traction surfaces such as wet asphalt, compacted snow, and polished ice. The autonomous system was evaluated across a diverse suite of scenarios specifically designed to challenge its planning and responsiveness under adverse conditions. These included high-speed cornering on snow-covered roads, dynamic adaptation to sudden ice patches, sharp emergency braking, and evasive maneuvers to avoid virtual obstacles. These trajectories were incorporated into the test dataset for evaluating dynamic modeling performance.

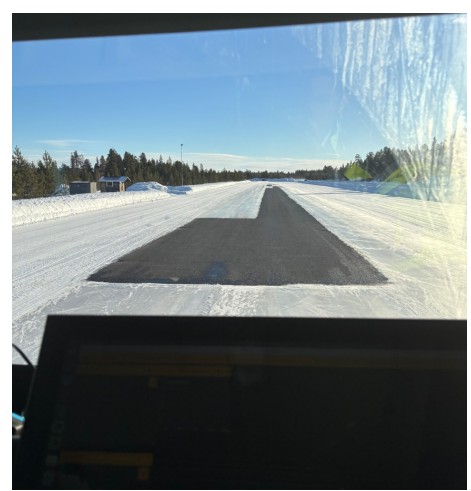

Figure 7: Mu-split test scenario captured from within the vehicle cab, illustrating asymmetric surface conditions—one side of the vehicle traveling on high-friction asphalt while the other traverses low-friction polished ice.

A particularly critical set of experiments was performed on a mu-split track (see Fig. 7), where one side of the vehicle experienced high-friction dry asphalt while the opposite side traversed low-friction ice. This asymmetric traction environment is known to induce significant yaw moments, increasing the risk of vehicle instability, trailer swing, and potential jackknifing. These mu-split tests constituted a critical stress evaluation of the model's ability to generalize under distributional shift, revealing key aspects of its stability and robustness. They also played a central role in validating the planner's strategies for maintaining safe and dynamically feasible behavior across varying surface conditions and friction transitions.

For data processing, raw trajectory data were preprocessed to enhance signal fidelity and facilitate effective model learning. A fourth-order Butterworth low-pass filter with a 5 Hz cutoff frequency was

applied to suppress high-frequency noise in the state signals. To ensure balanced feature scaling across the neural network, all input features were normalized to the range [–1, 1]. Each system state was encoded as a 7-dimensional vector comprising longitudinal and lateral velocities and accelerations, the tractor's yaw angle, the articulation angle between the tractor and trailer, and the front-wheel slip angle. The control input vector included throttle, brake, and steering commands. The resulting dataset contains over 10 hours of curated trajectory data. To promote diversity in dynamic behaviors, straight-line highway driving segments—which were overrepresented in raw data—were limited to 40% of the training set. This ensured sufficient representation of more complex scenarios, including turns, braking events, and evasive maneuvers.

# C Additional Results

**Base Simulation Environment.** The environments used in our experiments encompass a broad spectrum of dynamical behaviors and distribution shifts. As shown in Fig. 8, they include both robotic manipulation and locomotion tasks—ranging from the Panda-Lift setup in the RoboSuite environment to established continuous control benchmarks such as Ant, HalfCheetah, Hopper, and Walker.

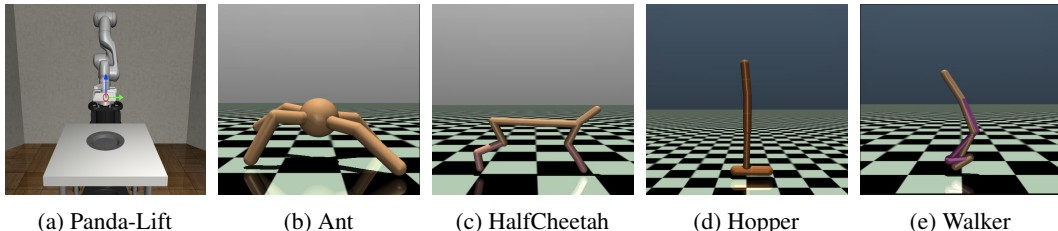

(a) Panda-Lift     (b) Ant     (c) HalfCheetah     (d) Hopper     (e) Walker

Figure 8: Environments: Panda-Lift, Ant, HalfCheetah, Hopper, and Walker.

## C.1 Predictive Performance

We extend our evaluation to four additional environments—Ant, HalfCheetah-Stationary, Panda, and Walker—to further test METAKOOPMAN's versatility. As shown in Table 2, the model consistently outperforms strong baselines across varied conditions.

Across all environments, METAKOOPMAN achieves the lowest validation loss as illustrated in Figure 9. In the **Ant** and **Walker** tasks, it rapidly adapts to changing dynamics and maintains stable performance under noise and contact variability. In the **HalfCheetah-Stationary** environment, it preserves robustness without overfitting in the absence of distribution shift. Finally, in the **Panda-Damping** manipulation setting, METAKOOPMAN attains the best overall accuracy with minimal variance, highlighting its scalability and generalization across locomotion and manipulation domains.

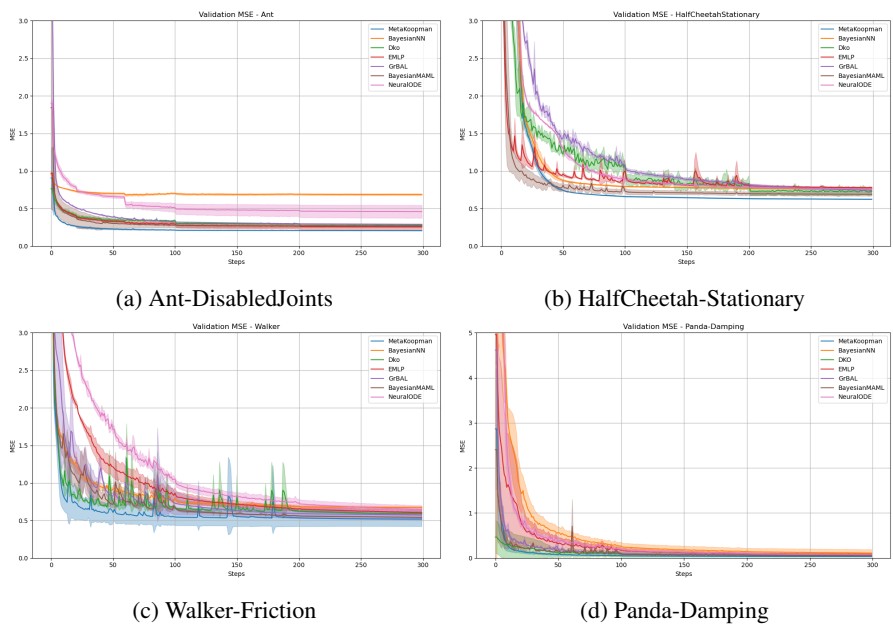

(a) Ant-DisabledJoints        (b) HalfCheetah-Stationary

(c) Walker-Friction        (d) Panda-Damping

Figure 9: Validation MSE over time for (a) Ant, (b) HalfCheetah-Stationary, (c) Walker, and (d) Panda-Damping. Shaded regions denote standard deviation across three runs.

Table 2: Final validation loss (± std) across additional environments. Best performance per environment is **bolded**.

| Environment | MetaKoopman | BayesianMAML | GrBAL | DKO | BayesianNN | EMLP | NeuralODE |
|---|---|---|---|---|---|---|---|
| **Ant-DisabledJoints** | **0.21 ± 0.00** | 0.27 ± 0.02 | 0.29 ± 0.00 | 0.27 ± 0.00 | 0.69 ± 0.01 | 0.26 ± 0.04 | 0.46 ± 0.09 |
| **HalfCheetah-Stationary** | **0.62 ± 0.00** | 0.69 ± 0.02 | 0.74 ± 0.02 | 0.73 ± 0.05 | 0.76 ± 0.00 | 0.78 ± 0.01 | 0.76 ± 0.02 |
| **Walker-Friction** | **0.52 ± 0.10** | 0.54 ± 0.00 | 0.57 ± 0.07 | 0.59 ± 0.00 | 0.67 ± 0.02 | 0.61 ± 0.03 | 0.64 ± 0.05 |
| **Panda-Damping** | **0.035 ± 0.002** | 0.05 ± 0.0002 | 0.07 ± 0.002 | 0.06 ± 0.003 | 0.11 ± 0.004 | 0.08 ± 0.02 | 0.07 ± 0.007 |

## C.2 Uncertainty Quantification.

We evaluate the quality of predictive uncertainty by measuring its correlation with actual prediction errors. A strong positive correlation indicates that the model expresses higher uncertainty when its predictions are less accurate—a desirable property for downstream safety-critical applications.

To assess this, we compute the Pearson correlation coefficient between the predicted uncertainty and mean squared error (MSE) across 100,000 randomly selected states spanning diverse trajectories and time steps. As shown in Table 3, METAKOOPMANconsistently exhibits the highest correlation values across all datasets, indicating well-calibrated uncertainty estimates.

Table 3: Correlation between predicted uncertainty and squared error across datasets. Higher values indicate better uncertainty calibration.

| Dataset | EMLP | BNN | Bayesian MAML | MetaKoopman |
|---|---|---|---|---|
| Truck Dataset | 0.58 | 0.48 | 0.56 | **0.68** |
| Ant-DisabledJoints | 0.59 | 0.53 | 0.58 | **0.72** |
| HalfCheetah-Slope | 0.63 | 0.52 | 0.62 | **0.69** |

# D Ablation Studies

We perform a series of ablation studies to assess the contribution of key design choices in METAKOOP-MAN. In particular, we investigate the role of the tempering mechanism in adaptation and evaluate the effect of varying history lengths on predictive performance. These experiments provide insight into the trade-offs between accuracy and computational efficiency, and help identify configurations that most effectively balance the two.

## D.1 Effect of Tempering

We evaluate the impact of Bayesian tempering by comparing METAKOOPMAN with and without this mechanism across three representative environments. As shown in Table 4, incorporating tempering consistently improves predictive accuracy, particularly in settings with pronounced distribution shifts or perturbations such as the truck dynamics and Ant environments.

Table 4: Performance with and without tempering (Avg. MSE).

| Environment | Without tempering | With tempering |
|---|---|---|
| Truck Dynamics | 0.084 | **0.059** |
| Walker-Friction | 0.56 | **0.52** |
| Ant-DisabledJoints | 0.235 | **0.21** |

## D.2   Effect of History Length

We investigate the effect of trajectory history length on predictive performance across three environments: Hopper-Gravity, HalfCheetah-Slope, and Truck Dynamics. The history length determines the temporal context available to the trajectory encoder, which can influence the model's ability to infer latent dynamics. Table 5 presents results for varying history lengths, revealing a consistent improvement in accuracy as more context is incorporated. Notably, longer histories yield the best performance, though with diminishing returns—highlighting some sort of trade-off between model expressiveness and computational efficiency.

Table 5: Effect of History Length on Prediction Accuracy (Mean Squared Error).

| Environment | 1 | 4 | 8 | 16 |
|---|---|---|---|---|
| Hopper-Gravity | 0.048 | 0.046 | 0.04 | **0.036** |
| HalfCheetah-Slope | 0.857 | 0.846 | 0.823 | **0.749** |
| Truck Dynamics | 0.0889 | 0.084 | 0.075 | **0.06** |

# E    Motion Planning with Cost-Guided Latent Action Sampling

We integrate METAKOOPMAN into a real-time motion planner for generating dynamically feasible trajectories in a truck–trailer system under complex physical constraints. To enable efficient planning at scale, we adopt a Cost-Guided Latent Action Sampling (CLAS) approach that leverages the learned variational action encoder to directly sample from the latent control space.

Instead of sampling actions from the original control space and encoding them at test time, CLAS samples directly from the latent Gaussian space learned by the variational encoder (Figure 10). Each sampled latent action is passed through the Koopman-based dynamics model via a small number of matrix multiplications to produce a candidate sub-trajectory over a fixed planning horizon.

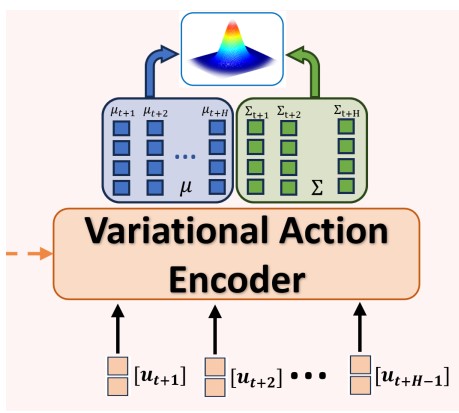

This approach yields significant gains in computational efficiency. Each trajectory rollout requires only a single forward pass through the context encoder, a Bayesian update of the Koopman prior, and a sequence of matrix multiplications that scale linearly with the number of sampled trajectories—eliminating the need for costly action encoding operations. Consequently, *CLAS* can generate and evaluate hundreds of thousands of dynamically consistent trajectories in parallel.

Figure 10: Variational action encoder used in the CLAS planner. The encoder learns a Gaussian distribution over action embeddings, enabling efficient sampling for trajectory rollout.

After obtaining the predicted trajectories, each candidate trajectory is scored according to a cost function that incorporates goal proximity, smoothness, and obstacle avoidance. The trajectory with the lowest overall cost is selected for execution. This planning formulation ensures that trajectories are not only feasible under the vehicle's dynamics, but also adapted to real-world factors such as low-traction surfaces, trailer articulation limits, and tight spatial constraints. The full planning procedure is detailed in Algorithm 2.

---

**Algorithm 2** Motion Planning with Cost-Guided Latent Action Sampling (CLAS)

---

**Require:** Initial state $s_0$, goal state $s_{\text{goal}}$, planning horizon $H$, number of trajectory samples $N$
**Ensure:** Dynamically feasible trajectory to goal
  1: Initialize empty set of trajectories $\mathcal{T} \leftarrow \emptyset$
  2: **for** $i = 1$ to $N$ **do**                ▷ This loop is fully parallelizable across samples
  3:      Sample a sequence of latent actions $\tilde{\mathbf{u}}^{(i)}_{1:H} \sim \mathcal{N}(0, I)$
  4:      Integrate dynamics: $s^{(i)}_{1:H} \leftarrow \texttt{KoopmanRollout}(s_0, \tilde{\mathbf{u}}^{(i)}_{1:H})$
  5:      Compute cost $J_i \leftarrow \texttt{EvaluateCost}(s^{(i)}_{1:H}, s_{\text{goal}})$
  6:      Store $(\tilde{\mathbf{u}}^{(i)}_{1:H}, s^{(i)}_{1:H}, J_i)$ in $\mathcal{T}$
  7: **end for**
  8: Select $\tilde{\mathbf{u}}^{*}_{1:H} \leftarrow \arg\min_{\tilde{\mathbf{u}}^{(i)}_{1:H}} J_i$ from $\mathcal{T}$
  9: **return** Trajectory corresponding to $\tilde{\mathbf{u}}^{*}$

---

The sampling and rollout steps (lines 3–6) are parallelizable across $N$ trajectories, allowing the method to leverage batched computation on GPUs or distributed hardware. This property enables CLAS to evaluate hundreds of thousands of dynamically consistent trajectories in parallel.

### E.1 Runtime Analysis

To highlight the efficiency improvements of latent action sampling, We evaluate the runtime efficiency of our method and baselines on the truck dataset, selected for its moderate complexity and relevance to online control. All experiments are conducted on an NVIDIA A10 GPU to ensure fair comparisons across models. Reported runtimes correspond to the inference phase only and are averaged over 1,000 evaluations to reduce variance from transient hardware effects.

Our method, leverages the variational action encoder to sample directly from the latent Gaussian space, avoiding the need for deterministic encoding of control inputs at test time. This modification yields substantial computational savings while maintaining predictive performance—making it especially well-suited for real-time planning scenarios.

Table 6 summarizes average per-step inference times for all compared models. Notably, METAKOOP-MAN W. VARIATIONAL ACTION ENCODER achieves the lowest runtime, outperforming both standard Koopman baselines and more complex probabilistic models such as Bayesian MAML, BNNs and ensembles by a wide margin.

Table 6: Inference Runtime Analysis (Average Time per Step in milliseconds).

| Method | Runtime (ms) |
|---|---|
| METAKOOPMAN | **0.087** |
| Deep Koopman Operator (DKO) | 0.21 |
| Bayesian Neural Networks (BNNs) | 1.35 |
| MC Dropout | 0.98 |
| Neural ODEs (NODE) | 0.22 |
| Ensemble MLP (EMLP) | 0.49 |
| GrBAL | 0.37 |
| Bayesian MAML | 0.64 |

# F  Experimental Details

## F.1  Simulation Environments

We evaluate METAKOOPMAN across five simulated control environments implemented using MuJoCo [39], each designed to assess the model's adaptability to both distributional shifts and structural perturbations. Distributional shifts include variations in gravity, terrain slope, and surface friction, while structural perturbations are introduced via randomized joint failures. To evaluate generalization beyond the training conditions, meta-training is performed over a constrained range of environmental parameters, and meta-testing spans a broader, previously unseen set of conditions, as detailed in Table 7.

Trajectory data is collected using a TD3 policy trained for one million steps on the full meta-test distribution of each environment. This policy is then used to generate trajectories for both meta-training and meta-testing phases. To promote diversity in the data, the agent follows its learned policy while incorporating stochastic action perturbations, where random actions are taken with a fixed probability. To simulate sensor noise, Gaussian observation noise with a standard deviation of $0.001$ is added in all environments.

| Environment | Meta-Train Range | Meta-Test / TD3 Range | Perturbation Type |
|---|---|---|---|
| HalfCheetah-Slope | $[-10°, 10°]$ | $[-15°, 15°]$ | Terrain slope |
| Walker2d-Friction | $[0.3, 3.0]$ scaling | $[0.2, 5.0]$ scaling | Surface friction |
| Hopper-Gravity | $[-1.0, 1.0]$ offset | $[-1.5, 1.5]$ offset | Gravitational field |
| Ant-DisabledJoints | Partial joint set | Full joint set | Actuator failure |

Table 7: Parameter ranges and shift types for simulated environments.

**HalfCheetah-Slope.**   This environment simulates locomotion on inclined terrain by varying the ground plane angle at the start of each episode, affecting both balance and propulsion.

**Walker2d-Friction.**   To model variability in ground contact, the friction coefficients of all ground-facing surfaces are scaled per episode.

**Hopper-Gravity.**   The vertical gravitational force is perturbed by offsetting the standard $-9.81$ value, influencing jump timing and overall dynamics.

**Ant-DisabledJoints.**   Mechanical degradation is simulated by disabling a randomly selected joint via nullification of its control signal. Failures are restricted to a subset of joints during meta-training, while meta-testing considers the full joint set.

**HalfCheetah-Stationary.**   This environment retains the standard `HalfCheetah-v5` configuration without any induced perturbations or distribution shifts. It serves as a stationary baseline to evaluate the robustness of meta-learned models in settings where the underlying dynamics remain consistent. This allows us to assess whether adaptation mechanisms degrade performance when no shift is present.

## F.2  Implementation Details.

This section provides implementation specifics for the proposed method, METAKOOPMAN, as well as concise summaries of the baseline methods used for empirical comparison. We describe architectural choices, adaptation strategies, and training hyperparameters. Common settings shared across experiments are listed at the end to avoid redundancy.

**Proposed Method: METAKOOPMAN.**   METAKOOPMAN integrates Koopman operator theory with Transformer-based sequence models to capture latent linear dynamics under distributional shift. The architecture employs a Transformer encoder-decoder structure, with the context encoder implemented as a standard Transformer encoder and the action encoder as a Transformer decoder. This configuration allows future action embeddings to attend to the historical context effectively.

Input sequences are constructed by concatenating state and action vectors over 16 past time steps, resulting in input tokens of dimension $n_x + n_a$, where $n_x$ and $n_a$ denote the state and action dimensions, respectively. These tokens are projected into a hidden dimension of 48 via linear layers. To encode temporal structure, we use Rotary Positional Embedding (RoPE) [36]. The encoder utilizes full self-attention, while the decoder employs a sliding-window causal attention mechanism with a window size of 8, enforcing autoregressive structure and limiting attention to recent context.

The decoder processes future action sequences, similarly projected into the hidden dimension. A learnable start token is prepended to the decoder input to initiate sequence generation. This design enables structured conditioning on both past trajectories and future controls, supporting closed-form Bayesian updates in latent space.

**Deep Koopman with Control (DKO).**    The DKO baseline models nonlinear dynamics by learning a latent linear system via separate encoders for states and control inputs. The state encoder maps raw states to a high-dimensional embedding, which is concatenated with the original state before further processing. It consists of fully connected layers with ReLU activations: [`State Dim` $\rightarrow 32 \rightarrow 64 \rightarrow 128 \rightarrow 84 \rightarrow$ `State Embedding Dim`] (ReLU applied to all but the output layer). The control encoder processes joint state-action inputs through a parallel MLP: [`State+Action Dim` $\rightarrow 32 \rightarrow 64 \rightarrow 128 \rightarrow 84 \rightarrow$ `Control Embedding Dim`]. The learned dynamics are governed by a linear Koopman operator, with system matrix $\mathbf{A}$ initialized from a Gaussian distribution and orthogonalized using singular value decomposition (SVD), and a control matrix $\mathbf{B}$ that maps control embeddings into the latent space.

**Neural Ordinary Differential Equation (Neural ODE).**    The Neural ODE baseline models continuous-time dynamics by learning the time derivative of the system state as a function of the current state and action. The ODE function $f_\theta(x, a)$ is parameterized by a fully connected neural network, which takes as input the concatenated state-action vector of dimension $n_x + n_a$. After that, the hidden layers of the network have the widths of: [64, 96, 128, 96, 84, 32], all using ReLU activations. The output layer produces a vector of dimension $n_x$, corresponding to the predicted time derivative of the state. Time integration is performed using a fourth-order Runge-Kutta solver from the `torchdiffeq` library.

**Ensemble Neural Networks.**    This baseline models system dynamics using an ensemble of ten feedforward networks. Each network takes a concatenated state-action input of dimension $n_x + n_a$ and passes it through a multilayer perceptron with hidden sizes [32, 64, 96, 64, 32], using ReLU activations throughout. The output is a state prediction of dimension $n_x$. Uncertainty is estimated via the variance of the ensemble predictions, capturing epistemic uncertainty arising from model disagreement.

**Bayesian Neural Network (BNN).**    The BNN baseline models predictive uncertainty by placing Gaussian distributions over the weights and biases of the network. The dynamics model is implemented using Bayesian linear layers, where each layer maintains a variational posterior over its parameters. The input, a concatenated state-action vector of dimension $n_x + n_a$, is passed through a multilayer architecture with hidden sizes [32, 64, 128, 84, 32], using ReLU activations, and outputs the predicted next state of dimension $n_x$. All priors are zero-mean unit-variance Gaussians.

The training objective combines mean squared error with a Kullback–Leibler divergence term to regularize the variational posterior:

$$\mathcal{L} = \mathcal{L}_{\text{data}} + \beta \, \text{KL}(\text{posterior} \parallel \text{prior}),$$

with $\beta = 10^{-5}$. During inference, ten forward passes using Monte Carlo sampling produce mean predictions and diagonal covariance estimates, enabling uncertainty-aware forecasting.

**Bayesian Model-Agnostic Meta-Learning (BMAML)**    The Bayesian MAML (BMAML) model extends gradient-based meta-learning by maintaining a particle-based ensemble to approximate the task posterior. Each particle represents a distinct instantiation of a shared dynamics model, parameterized as a feedforward neural network with layers [$(n_x + n_a) \rightarrow 144 \rightarrow 96 \rightarrow n_x$] and ReLU activations, where $n_x$ and $n_a$ are the dimensions of state and action inputs, respectively. The ensemble is initialized from a learned meta-prior and adapted per task using Stein Variational Gradient Descent (SVGD), which preserves uncertainty structure by coordinating updates across particles

using an RBF kernel. During inference, future state trajectories are predicted by averaging rollouts from all particles. The model outputs both the predictive mean and variance, capturing epistemic uncertainty in a manner suited for few-shot adaptation.

**Gradient-Based Adaptive Learner (GrBAL)**   The Gradient-Based Adaptive Learner (GrBAL) enables fast online adaptation of a neural dynamics model by leveraging meta-learned initial parameters optimized for fast adaptation. The model is a feedforward network with layers $[(n_x + n_a) \rightarrow 56 \rightarrow 108 \rightarrow 56 \rightarrow n_x]$, where $n_x$ and $n_a$ are the dimensions of the state and action respectively, and ReLU activations are used throughout. GrBAL performs inner-loop adaptation on each task by computing gradients of the prediction loss over recent trajectory segments and updating the model weights accordingly. This is implemented using differentiable inner-loop optimization with higher-order gradients, allowing backpropagation through the adaptation process. During inference, the adapted model predicts future state trajectories from the last observed state and planned action sequence. GrBAL is designed to support rapid and effective adaptation in dynamic environments, particularly when system dynamics change abruptly due to disturbances or unmodeled factors.

**Common Implementation Details.**   To ensure comparability, all models are configured to have approximately 60k trainable parameters, with the exception of Bayesian MAML and Ensemble Neural Networks, which consist of multiple networks and therefore contain significantly more parameters. All models are trained using the AdamW optimizer with a learning rate of $3 \times 10^{-3}$ and weight decay of $1 \times 10^{-2}$. A `StepLR` scheduler decays the learning rate by a factor of 0.3 at one-third intervals over the course of 300 training epochs. All models are trained with a batch size of 128 and a fixed prediction horizon of 32 future time steps. Meta-learning models are provided with an additional context window of 16 time steps for adaptation. The primary training objective is the Mean Squared Error (MSE) between predicted and ground truth future states, except for METAKOOPMAN, which minimizes the negative log-likelihood (NLL). Gradient clipping with a maximum norm of 1.0 is applied to all feedforward networks to improve training stability. A summary of baseline capabilities with respect to adaptation, uncertainty estimation, and structured dynamics modeling is provided in Table 8.

| Method | Adaptation | Uncertainty | Structured Dynamics |
|---|:---:|:---:|:---:|
| METAKOOPMAN | ✓ | ✓ | ✓ |
| Bayesian MAML | ✓ | ✓ | |
| GrBAL | ✓ | | |
| Deep Koopman (DKO) | | | ✓ |
| Ensemble NN | | ✓ | |
| Bayesian NN | | ✓ | |
| Neural ODE | | | ✓ |

Table 8: Comparison of baseline capabilities in terms of adaptation, uncertainty quantification, and incorporation of structured dynamics.

