# OpenReview forum: "MetaKoopman: Bayesian Meta-Learning of Koopman Operators for Modeling Structured Dynamics under Distribution Shifts"
_NeurIPS.cc/2025/Conference — NeurIPS 2025 poster_

### Official Review · Reviewer_cqV4 · 2025-07-01

**Clarity:** 3
**Significance:** 2
**Originality:** 2
**Rating:** 3
**Confidence:** 3

**Summary:**

This paper proposes MetaKoopman, a framework that combines Koopman operator theory, Bayesian inference, and meta-learning for dynamics modeling under distribution shift. The approach enables fast adaptation and uncertainty-aware forecasting, with evaluation on simulated environments and a real-world autonomous truck dataset.

**Questions:**

1. What is the purpose of Lemma 4.1 and 4.2 in the context of the paper? Are they supporting a particular theoretical claim? It is unclear whether formalizing them as lemmas is necessary.

2. The real-world dataset is described as being collected across a wide variety of scenarios (e.g., snow, ice, mixed-friction). While this is valuable, it raises concerns about whether the test-time scenarios are actually out-of-distribution. The meta-learner might act more like a state switcher than a generalization engine. How do the authors ensure that test conditions are truly novel and that the adaptation mechanism is not simply memorizing specific dynamics from training?

3. In Table 1, the metric shown is referred to as "validation loss." Could the authors clarify what "validation" means in this context? Why is it not test error?

4. In Line 302 the text mentions that "Transformer-based embedding function" is kept fixed after training. However, there is no mention of a Transformer model in the section 4.

5. No hyperparameter sensitivity analysis.

**Ethical Concerns:**

["NO or VERY MINOR ethics concerns only"]

**Final Justification:**

While the method is well presented, more real-world datasets and comparative methods with distribution shifts are needed for validation. The current version still requires additional work. I am inclined to keep my current score.

**Limitations:**

yes

**Quality:**

2

**Strengths And Weaknesses:**

Strengths:
1. The proposed system demonstrates clearly performance on the experimental datasets.

2. The authors collect a long-duration, multi-scenario real-world dataset from a full-scale autonomous truck system and evaluate the method under realistic operational conditions. This greatly enhances the practical relevance of the work. If released, the dataset would also be a valuable contribution to the community.

3. The paper is well-written and generally easy to follow, with a clear explanation of the system design.

Weaknesses:
1. While the combination of Koopman operators, Bayesian inference, and meta-learning is technically sound, the framework feels loosely assembled. The motivation and necessity for integrating these specific components is not clearly justified. Each module appears replaceable. For instance, one could use an ODE, SDE, or a Transformer instead of the Koopman operator, or HyperNetworks in place of meta-learning. As such, the novelty is somewhat marginal and the contribution seems incremental.

2. The paper lacks a thorough review and discussion of existing work on handling time-series distribution shifts. Numerous relevant methods have been proposed in meta-learning, continual learning, test-time adaptation, and uncertainty-aware modeling, yet the paper does not engage with this literature, nor are they included in the comparisons.

3. The evaluation is limited in both baselines and datasets. As said before, several strong methods in the distribution shift literature are missing in the baselines. Additionally, the only real-world dataset used is from a single truck platform, with no validation on datasets from different domains or modalities. This limits the generality and persuasiveness of the claims.

4. Figure 1 is not well integrated into the text. Its visual design is not intuitive, making it difficult to understand the proposed workflow. Figure 3 presents the comparison in a visually entangled way, making it hard to discern differences among methods.

---

> ### Author Rebuttal · Authors · 2025-07-31
>
> We thank the reviewer for their constructive feedback. We are encouraged by the positive remarks regarding the clarity, real-world dataset, and system design. We address the concerns raised below:
>
> ---
>
> > * **W1: While the combination of Koopman operators, Bayesian inference, and meta-learning is technically sound, the framework feels loosely assembled...**
>
> We thank the reviewer and would like to clarify the motivation of our design. Our objective is to perform real-time planning in safety-critical real-world conditions. This means we require an efficient model that is uncertainty-aware and can adapt to distribution shifts.
>
> * Koopman operators offer a way of modeling nonlinear dynamics via linear evolution in a latent space, enabling efficient long-horizon prediction. Importantly, Transformers are not an alternative to Koopman modeling; In fact, in our case, they are used to learn temporally informed latent embeddings where Koopman dynamics apply.
>
> * Regarding ODE/SDE-based alternatives, we agree they are valuable. However, their dependence on numerical solvers often leads to inefficiencies and challenges in uncertainty propagation. As shown in Table 1, 2, MetaKoopman consistently outperforms Neural ODEs. We also evaluated against physics-based ODE models (not included in the paper for space and scope reasons), which we are happy to share upon request.
>
> * Bayesian inference enables closed-form updates over the Koopman operator, directly modeling dynamics uncertainty — a capability absent in prior Koopman methods that used point-estimate operators. To our knowledge, this is the first tractable probabilistic Koopman with explicit dynamics uncertainty quantification.
>
> * Meta-learning is critical for adaptation. While hypernetworks are one class of meta-learners, they are typically computationally intensive and do not easily support uncertainty quantification. In contrast, our approach meta-learns a conjugate prior, enabling fast updates without test-time optimization—a key requirement for planning/control.
>
> Each component in MetaKoopman addresses a specific modeling challenge: Koopman for structure and prediction, Bayesian inference for uncertainty quantification, and meta-learning for adaptation.
>
> ---
>
> > * **W2: The paper lacks a thorough review and discussion of existing work on handling time-series distribution shifts… the paper does not engage with this literature, nor are they included in the comparisons.**
>
> Thank you for bringing up time-series distribution shifts. We focused on literature related to controllable systems of the form $f(x(t),u(t))$ instead of systems of the form $f(t)$ since they have more complex nuances. However, we agree that some of this work may offer complementary perspectives and will include a brief discussion in the revised version.
>
> On the meta-learning side, we agree that it is essential. In Section 2, we discuss several meta and bayesian meta-learning methods, highlighting that MetaKoopman differs by offering closed-form adaptation and principled uncertainty quantification. Our evaluation (Tables 1–3, 6) includes both meta-learners and probabilistic baselines (e.g., BNNs, ensembles) to ensure a comprehensive comparison.
>
> For continual learning, our setup differs: we model piecewise-stationary regimes with rapid shifts (e.g., changes in surface friction), rather than continuous data streams. Nonetheless, we acknowledge the relevance and will include a discussion on this distinction.
>
> ---
>
> > * **W3: The evaluation is limited in baselines and datasets...the only real-world dataset used is from a truck platform, with no validation from different domains or modalities. This limits the generality and persuasiveness of the claims.**
>
> Thank you for raising this point. The real world dataset presented in the paper is one of the world's first demonstrations of autonomous driving with heavy trucks on ice. We believe this is a significant contribution to embodied AI.
>
> Our experiments cover **seven environments** and a diverse set of distribution shifts: physical (e.g., gravity in *Hopper-Gravity*, slope in *HalfCheetah-Slope*, friction in *Walker-Friction*), structural (e.g., joint failures in *Ant-DisabledJoints*), and real-world non-stationary settings (e.g., snow, ice, and mu-split in the truck dataset). We also include a new robotic manipulation task under varying damping conditions. The results for the manipulation is shown below, highlighting MetaKoopman's strong generalization across **driving, locomotion, and manipulation** domains.
>
> |**metakoopman**|**bmaml**|**grbal**|**DKO**|**BNN**|**EMLP**|**NODE**|
> |-|-|-|-|-|-|-|
> |**0.035±.001**|0.05±.002|0.07±.004|0.06±.002|0.11±.004|0.08±.012|0.07±.008|
>
> **Table 1** Prediction error (averaged over 3 seeds) across 32 timesteps.
>
> Regarding baselines, we compare against **six widely used and representative methods**:
>
> - **Bayesian MAML** (meta-learning and uncertainty),
> - **GrBAL** (meta-learning),
> - **Deep Koopman** (structure-based),
> - **Neural ODEs** (continuous-time modeling),
> - **Bayesian NNs, Ensemble models** (uncertainty).
>
> **MetaKoopman** outperforms these baselines in terms of **prediction accuracy** (Tables 1–2), **uncertainty calibration** (Table 3), **inference speed** (Table 6), and **safety-critical planning outcomes** (Section 5.2, Figs. 4–5). A summary of comparative capabilities is provided in **Table 8**.
>
> That said, we agree broader validation is important and would be glad to add more experiments should the reviewer have specific suggestions.
>
>
>
>
> ---
>
> > * **W4: Figure 1 is not well integrated into the text... Figure 3 presents the comparison in a visually entangled way**
>
> We appreciate the reviewer’s feedback on Figures 1 and 3 and welcome the opportunity to clarify their intent:
>
> **Figure 1** provides a high-level overview of the *MetaKoopman* pipeline:
>
> 1. **Latent context encoding** maps nonlinear state-action trajectories into a latent space where dynamics evolve linearly;
> 2. **Closed-form Bayesian inference** adapts a distribution over the Koopman operator using recent trajectory data — our central contribution;
> 3. **Uncertainty-aware prediction** uses this adapted distribution to forecast future states with calibrated uncertainty.
>
> This sequence is described in Section 4 and visualized step-by-step in the figure (p. 4). We will revise the figure to improve clarity, as recommended.
>
> **Figure 3** illustrates the impact of adaptation on predictive accuracy using the truck dataset. It compares the prediction error **before and after** the update to the Koopman operator. As shown, the error decreases from **1.37 to 0.06**, validating the effectiveness of the adaptation. We will make Figure 3 easier to follow, as recommended.
>
> We thank the reviewer for the suggestions and are confident that the revised figures will improve readability.
>
>
>
> ---
> ---
>
> **We now answer the reviewer's questions:**
>
> > * **Q1: What is the purpose of Lemma 4.1 and 4.2 in the context of the paper? Are they supporting a particular theoretical claim? It is unclear whether formalizing them as lemmas is necessary.**
>
> Thank you for raising this question. **Lemma 4.1 and Lemma 4.2 form the theoretical foundation of the closed-form Bayesian inference procedure at the core of MetaKoopman.**
>
> - **Lemma 4.1** shows that under a MNIW prior, the posterior over $\mathbf{K}$ and $\boldsymbol{\Sigma}$ remains in the same conjugate family, enabling fast, analytic updates at test time—without gradient-based optimization like prior works.
>
> - **Lemma 4.2** derives the posterior predictive over future latent states, supporting uncertainty-aware multi-step forecasting and planning.
>
> We present these as formal lemmas to emphasize their foundational role in our framework.
>
> ---
>
> > * **Q2: The real-world dataset is collected across a variety of scenarios...How do the authors ensure that test conditions are novel and that the adaptation is not memorizing dynamics from training?**
>
> We appreciate this question. Each task is a trajectory segment reflecting latent conditions (e.g., surface, slope) that evolve unpredictably over time. We note that challenging scenarios (including both the polished ice and mu-split (Appendix Fig. 7)) were only used in the test dataset, to make sure the model did not learn these nuances but instead adapted to them.
>
> ---
>
> > * **Q3: In Table 1, the metric shown is referred to as ‘validation loss.’ Could the authors clarify what ‘validation’ means? Why is it not test error?**
>
> Thank you for the close reading. The metric in Table 1 reflects the final test loss on held-out trajectories. We agree that using "validation" was misleading and will revise the terminology for clarity in the updated version.
>
> ---
>
> > * **Q4: In Line 302 the text mentions that 'Transformer-based embedding function' is kept fixed. However, there is no mention of a Transformer in Section 4.**
>
> Thank you for raising this point. Both the context and action encoders use Transformer architectures, allowing them to model delayed state-action sequences. While Section 4 describes the framework in a model-agnostic way, full architectural details are provided in Appendix F.2. This is also referenced at the end of Section 4.1 (Line 303). We agree this could be made clearer and will revise the main text to improve clarity.
>
> ---
>
> > * **Q5: No hyperparameter sensitivity analysis.**
>
> Thank you for raising this point. Appendix D (Table 5) presents a sensitivity analysis on history length, showing consistent gains up to a saturation point. Table 4 ablates the tempering mechanism, where meta-learned tempering improves performance under distribution shift. We’re happy to include further ablations upon request and will add more results in the next version.
>
> ---
>
> Thank you for the thoughtful feedback. We hope our responses have addressed the main concerns and clarified our contributions. We're happy to provide any clarification or answer any further questions the reviewer might have.

---

> > ### Comment · Reviewer_cqV4 · 2025-08-04
> >
> > Thank you for the response.
> >
> > The clarifications provided have helped make certain aspects of the paper clearer.
> >
> > However, my two main concerns remain.
> > 1) While the proposed method is intended to be general-purpose, the current experimental setup makes it appear task-specific. This weakens the perceived generality and broader applicability of the approach. Incorporating more diverse real-world datasets across different domains and modalities would greatly strengthen the empirical support for its broad applicability.
> >
> > 2) The comparison with existing methods for handling distribution shifts remains limited. Without stronger baselines from recent literature on distribution shift, it is difficult to assess the method’s advantage over the current state of the art.

---

> > > ### Author Response · Authors · 2025-08-07
> > >
> > > We thank the reviewer for the feedback. Below we elaborate more on the concerns raised:
> > >
> > > > **While the proposed method is intended to be general-purpose, the current experimental setup makes it appear task-specific. Incorporating more diverse real-world datasets across different domains and modalities would greatly strengthen the empirical support for its broad applicability.**
> > >
> > > We thank the reviewer for the feedback and would like to clarify that the evaluation is both broad and grounded in a meaningful real-world deployment.
> > >
> > > - **Broad evaluation datasets:** The paper **covers 7 environments across 3 modalities**—autonomous driving, locomotion, and manipulation—under three shift types:
> > >   - *Real-world, non-stationary:* heavy-truck driving on snow, polished ice, and μ-split surfaces.
> > >   - *Physical-parameter shifts:* gravity (Hopper-Gravity), slope (HalfCheetah-Slope), friction (Walker-Friction), damping (Panda-Damping).
> > >   - *Structural faults:* joint failures (Ant-DisabledJoints).
> > >
> > > - **Real-World Deployment.** MetaKoopman was deployed on a real self-driving truck and trailer system under **challenging conditions, including polished snow, μ-split, and mixed asphalt–ice surfaces**. As shown in the paper and accompanying video, it enabled safe planning in previously unseen scenarios, demonstrating robustness to real-world distribution shifts.
> > >
> > > - **New cross-modality evidence.** During the rebuttal, we added the Panda-Damping manipulation task, and MetaKoopman remained state-of-the-art, underscoring generality.
> > >
> > > - **Depth of real-world data.** The truck dataset was collected over a full year on public roads and specialized test tracks to capture multiple weather- and friction-induced shifts. Replicating that level of diversity on an additional platform involves significant logistical and financial overhead.
> > >
> > > We would be happy to include additional datasets or evaluations if the reviewer has any specific recommendations.
> > >
> > > ---
> > >
> > > > **The comparison with existing methods for handling distribution shifts remains limited.**
> > >
> > > We thank the reviewer for the feedback. We compare against **six widely used and representative baselines**, including meta-learned approaches such as GrBAL[1] and Bayesian MAML[2], along with other methods based on the Koopman structure, continuous-time modeling, and uncertainty estimation. These baselines were chosen to reflect key directions in the literature on learning under distribution shift. If the reviewer has a specific reference in mind, we would appreciate that since we can be specific in addressing that concern.
> > >
> > > ---
> > > ###### [1] Nagabandi, A., Clavera, I., Liu, S., Fearing, R. S., Abbeel, P., Levine, S., & Finn, C. (2018). Learning to adapt in dynamic, real-world environments through meta-reinforcement learning. arXiv preprint arXiv:1803.11347.
> > >
> > > ###### [2]: Yoon, Jaesik, et al. "Bayesian model-agnostic meta-learning." Advances in neural information processing systems 31.

---

### Official Review · Reviewer_wjtk · 2025-07-03

**Clarity:** 3
**Significance:** 3
**Originality:** 3
**Rating:** 4
**Confidence:** 3

**Summary:**

This paper presents MetaKoopman, a novel Bayesian meta-learning framework for modeling nonlinear dynamical systems under distribution shift. The core idea is to learn a Matrix Normal-Inverse Wishart (MNIW) prior over the Koopman operator, enabling closed-form Bayesian inference for fast adaptation using short trajectory segments. The model is evaluated on both simulated environments (MuJoCo) and real-world autonomous truck trials in adverse winter conditions, showing strong improvements in prediction accuracy, uncertainty calibration, and robustness to distribution shift.

**Questions:**

1. Have the authors considered using a probabilistic context encoder or ensemble-based embeddings?
2. Can MetaKoopman handle high-dimensional control inputs or sparse rewards in reinforcement learning settings?

**Ethical Concerns:**

["NO or VERY MINOR ethics concerns only"]

**Final Justification:**

This work has solid theory and proof. My concerns have also been addressed properly. I keep my positive rating.

**Limitations:**

yes

**Quality:**

3

**Strengths And Weaknesses:**

Strengths
1. The method combines structured Koopman modeling with closed-form Bayesian updates, avoiding costly inner-loop optimization at test time. Clear derivation and use of conjugate priors (MNIW) allow efficient inference and uncertainty estimation.
2. While Koopman-based models have been used before, this work is the first to meta-learn a prior over them, addressing dynamic environments and shifts.
3. Contributions are well broken down: Bayesian operator, variational action encoder, large-scale dataset, etc.

Weaknesses
1. Although the truck setting is compelling, it remains unclear how well this method generalizes to higher-dimensional robotic tasks (e.g., manipulation, aerial vehicles).
2. MSE and closed-loop safety in two scenarios are useful, but the breadth of downstream planning metrics (e.g., success rate, collisions, time to goal) is not fully explored.

---

> ### Author Rebuttal · Authors · 2025-07-31
>
> We sincerely thank the reviewer for their thoughtful and constructive feedback. We appreciate the recognition of the novelty in meta-learning a prior over the Koopman operator, as well as the value of the closed-form updates for efficient inference and principled uncertainty quantification. Below, we address the reviewer’s concerns and questions in detail.
>
> ---
>
> > * **W1: Although the truck setting is compelling, it remains unclear how well this method generalizes to higher-dimensional robotic tasks (e.g., manipulation, aerial vehicles).**
>
> Thank you for highlighting this important point. While our real-world deployment focuses on a truck–trailer system, the MetaKoopman framework is designed to be domain-agnostic and scalable to high-dimensional systems. Our experiments were deliberately designed to test this scalability.
>
> In the experiments section of the paper, we include results on Ant, a high-dimensional 8-DoF agent with 113-dimensional observations and complex contact dynamics. As shown in Table 2 (Appendix C.1), MetaKoopman achieves the lowest error across all baselines.
>
> Motivated by the reviewer’s suggestion, we have further extended our evaluation to include a simulated 7-DoF robotic manipulation task under varying damping conditions—**Panda Lift** from *robosuite* [1]. Our evaluation now comprises a real-world autonomous driving dataset alongside six simulated environments, covering three distinct application modalities: self-driving, locomotion, and manipulation. The results for the manipulation task are presented below.
>
> | **metakoopman** | **bmaml**    | **grbal**     | **DKO**       | **BNN**       | **EMLP**     | **NODE**      |
> |-----------------|--------------|---------------|---------------|---------------|--------------|---------------|
> | **0.035 ± .001** | 0.05 ± .002  | 0.07 ± .004   | 0.06 ± .002   | 0.11 ± .004   | 0.08 ± .012  | 0.07 ± .008   |
>
> **Table 1** Prediction error (averaged over 3 seeds) across 32 timesteps. **MetaKoopman** continues to outperform prior methods, demonstrating its scalability and strong generalization capabilities beyond the locomotion and driving domains.
>
>
> We thank the reviewer for their excellent suggestion to enhance our evaluation, and we remain open to addressing any further comments or concerns they may have.
>
>
> ---
>
> > * **W2: MSE and closed-loop safety in two scenarios are useful, but the breadth of downstream planning metrics (e.g., success rate, collisions, time to goal) is not fully explored.**
>
> We agree that evaluating a broader set of downstream planning metrics would further strengthen the empirical case for our method. In our submission, we focused on two safety-critical real-world scenarios—adaptive braking on ice and evasive lane change on snow—where predictive accuracy under distribution shift directly impacts feasibility and safety.
>
> To that end, we conducted five independent full-scale trials per scenario. MetaKoopman succeeded in all five, while non-adaptive baselines failed to stop safely or misjudged braking feasibility. Although these outcomes are consistent, we acknowledge that this sample size may be small to establish statistical significance. Scaling up these tests is constrained by the cost and logistics of deploying a 37.5-ton vehicle under controlled winter conditions. We are happy to include these results in the paper if the reviewer thinks so. Additionally, we are open to conducting an extra reinforcement learning experiment to include further downstream metrics in the camera-ready version, should the reviewer recommends it.
>
>
> ---
> ---
>
> Below, we answer the reviewer's questions:
> > * **Q1: Have the authors considered using a probabilistic context encoder or ensemble-based embeddings?**
>
> We appreciate the reviewer raising this point. Yes, we have considered using a probabilistic context encoder, and we explicitly discuss this in Section 6 (Discussion) and Appendix A.1 (lines 797–805). However, motivated by our application, we aim to develop systems that are both rapidly adaptable and explicitly aware of uncertainty in the dynamics. For this reason, we chose to focus uncertainty modeling on the Koopman operator rather than the encoder.
>
> This modeling choice is guided by the fact that, in practice—especially under distribution shift—**uncertainty in the system dynamics dominates**. When test-time conditions deviate (e.g., due to changes in surface friction or payload), it is typically the dynamics—not the representation—that drive prediction quality. In our setting, the encoder is trained across a diverse task distribution, and its embeddings are generally stable and informative. Focusing uncertainty on the Koopman operator therefore offers the greatest benefit for both performance and calibration, while maintaining real-time efficiency.
>
> Nonetheless, we agree that extending MetaKoopman with distributional encoders (e.g., [2, 3, 3]) is a promising direction, particularly for regimes with partial observability or severe data sparsity, and we plan to explore this in future work.
>
> ---
>
> > * **Q2: Can MetaKoopman handle high-dimensional control inputs or sparse rewards in reinforcement learning settings?**
>
> Yes, similar to our discussion in Weakness 1 — where we evaluated MetaKoopman on high-dimensional tasks such as Ant and HalfCheetah — we believe that the approach can be extended to handle high-dimensional control inputs as well.
>
> Regarding sparse rewards, while our primary focus is on dynamics modeling and planning, we believe MetaKoopman can be seamlessly integrated into model-based reinforcement learning frameworks. Its ability to adapt quickly and model uncertainty in dynamics makes it especially promising for model-based and meta-RL methods.
>
>
> ---
>
> We sincerely thank the reviewer for their valuable feedback. We hope that our responses have adequately addressed the concerns raised and helped clarify the key points. We would be more than happy to provide further elaboration should any questions remain.
>
> ---
>
> ##### [1] Zhu, Yuke, et al. "robosuite: A modular simulation framework and benchmark for robot learning." arXiv preprint arXiv:2009.12293 (2020).
>
> ##### [2] Fraccaro, Marco, et al. "A disentangled recognition and nonlinear dynamics model for unsupervised learning." Advances in neural information processing systems 30 (2017).
>
> ##### [3] Yildiz, Cagatay, Markus Heinonen, and Harri Lahdesmaki. "Ode2vae: Deep generative second order odes with bayesian neural networks." Advances in Neural Information Processing Systems 32 (2019).
>
> ##### [4] Han, Minghao, Jacob Euler-Rolle, and Robert K. Katzschmann. "DeSKO: Stability-assured robust control with a deep stochastic Koopman operator." International conference on learning representations (ICLR). 2022.

---

> > ### Comment · Reviewer_wjtk · 2025-08-05
> >
> > Thanks for the rebuttal efforts made by the authors.
> >
> > Most of my concerns have been properly addressed. I will keep my score.

---

### Official Review · Reviewer_azPt · 2025-07-03

**Clarity:** 3
**Significance:** 3
**Originality:** 3
**Rating:** 5
**Confidence:** 4

**Summary:**

This paper introduces a hierarchical Bayesian framework for modelling dynamical systems via Koopman operators under distributional shift. Koopman operators model state transitions of dynamical systems by elevating the state-action space to a higher dimensional feature space where the dynamics are linear and can be effectively approximated by matrices after truncation. Instead of learning a single Koopman operator model, the paper focuses on learning a computationally efficient probability distribution over Koopman operators conditioned on trajectory data. Once learned, the model can be adapted online to changes in the trajectory distribution due to, e.g., changes in the terrain a vehicle is driving over. The paper proposes parameterising a prior over Koopman operators via a matrix normal inverse Wishart (MNIW) distribution, so that the posterior distribution is conjugate and available in closed form. States and actions are encoded via a context encoder, which maps state-action sequences to the sequences of observables (feature space vectors), capturing temporal dependences between steps. The method can also be used for multi-step prediction via an action encoder for the sequences of control/action signals. Experiments involving a range of synthetic and real data are presented, including evaluation on the control of a real-world autonomous truck driving in icy road conditions.

**Questions:**

* How are the parameters of the encoders learned? Are they learned within the same loop in Algorithm 1 or pre-trained via a separate process?

* How is the learning objective for the variational action encoder formulated?

**Ethical Concerns:**

["NO or VERY MINOR ethics concerns only"]

**Final Justification:**

My main concerns have been addressed, and I believe the feedback from the reviews and discussions in the rebuttal phase should significantly strengthen the paper if appropriately taken into account in the paper's revision.

**Limitations:**

The main limitations are discussed in Section 6.

**Quality:**

3

**Strengths And Weaknesses:**

### Strengths
* The paper addresses an important problem in the modelling of dynamical systems deployed in real-world situations.
* Comprehensive experimental evaluation against relevant baselines in simulated RL benchmark environments and on real truck trajectory data
* Real-world deployment on an autonomous truck-trailer system using the proposed method for uncertainty-aware motion planning
* Significant performance improvements with respect to traditional baselines

### Weaknesses
* **Missing related work.** The paper by Singh et al. (2025, available on ArXiv since 2024, reference below) seems to be addressing the same problem of adaptation of Koopman operators for robust control, though with a different technique. It'd be worth discussing it in the related work section. In addition, the current related work section focuses on Koopman operators and meta-learning for control, missing the whole body of literature on *robust control*, which the current framework could be fit within. Such discussion would help to contextualise the method and discuss the potential advantages/disadvantages of using Koopman-based frameworks for robustness.

* Some technical details are missing, which I elaborate further below.
    - The predictive distribution in Eq. 13 (Lemma 4.2) should be conditioned on the Koopman operator $\mathcal{K}$, which is treated as a random variable by the proposed method.
    - It is also not clear what the superscript means in Eq. 14, though it can be inferred from context that it indexes Koopman operator samples, which again are not explicitly conditioned on, over a rolling window.
    - I missed an explanation of how the context encoder and action encoder parameters are learned, whether it happens within Algorithm 1's loop or via separate pre-training.
    - The distribution of trajectories $\rho(\mathcal{T})$ is supposedly sampling trajectories under "distinct operating conditions", which is a bit confusing. It'd make sense to treat the operating conditions as random variables as well, in case it is a requirement that trajectories are sampled from different environments. Otherwise, in principle, a trajectories distribution may represent trajectories from the same environment.
    - The learning objective for the variational action encoder, one of the claimed contributions, should be stated in the main text, which currently only discusses its KL regularisation.

* As a minor adjustment, above Eq. 15, it should state the meta-objective is to minimise the *negative* log-likelihood (or maximise the log-likelihood) to match the objective described in the equation.

#### References
* Singh R, Sah CK, Keshavan J. Adaptive Koopman embedding for robust control of nonlinear dynamical systems. *The International Journal of Robotics Research*. 2025;0(0). doi:10.1177/02783649251341907 (ArXiv: https://arxiv.org/abs/2405.09101)

---

> ### Author Rebuttal · Authors · 2025-07-31
>
> We thank the reviewer for the thoughtful and constructive feedback. We appreciate their recognition of our work in tackling real-world modeling under distribution shifts, conducting comprehensive evaluations across synthetic and real benchmarks, demonstrating strong empirical gains over baselines, and deploying the approach on an autonomous truck. We also value their engagement with both theoretical and practical aspects of the work. Below, we address the reviewer’s comments and clarify several technical points.
>
> ---
>
> > ### **W1: Missing Related Work (Singh et al., 2025) [1]**
>
> We thank the reviewer for pointing us to the relevant and important work by Singh et al. (2025), which addresses online adaptation of Koopman models in the context of robust control. We agree that it is closely related and will incorporate a discussion of it in the final version.
>
> While both approaches tackle the broader challenge of adapting Koopman-based models to shifting system dynamics, our method differs in several aspects:
>
> - **Bayesian adaptation:** Singh et al. introduce an online neural correction module that learns residual dynamics. In contrast, our MetaKoopman model performs closed-form Bayesian updates to the Koopman operator using a meta-learned Matrix Normal–Inverse Wishart prior, enabling fast and analytically grounded adaptation.
>
> - **Uncertainty quantification:** Singh et al. do not explicitly model uncertainty. However, our MetaKoopman models predictive uncertainty by producing a posterior predictive distribution, which captures both epistemic and aleatoric uncertainty—important for decision-making under dynamic uncertainty, and not explicitly modeled in Singh et al.
>
> - **Meta-learning framework:** Rather than training an online residual module from scratch, we meta-learn a prior over Koopman operators across a distribution of tasks. This supports rapid adaptation from a few trajectory points without iterative optimization.
>
> We thank the reviewer for highlighting the relevant related work by Singh et al. We will update our manuscript to link their work to ours, and include a discussion of robust control methodologies in the related works section.
>
> ---
>
>
> > ### **W2: Some technical details are missing, which I elaborate further below.**
> >
> > * **The predictive distribution in Eq. 13 (Lemma 4.2) should be conditioned on the Koopman operator $\mathcal{K}$ which is a random variable.**
>
> Thank you for this insightful comment. You are correct that $\mathcal{K}$ is treated as a random variable in our formulation. As shown in Eq.18 of the appendix, the posterior predictive distribution is obtained by marginalizing over both $\mathcal{K}$ and $\Sigma$. This marginalization naturally leads to the posterior mean $\mathcal{M}$, as reflected in Eq.26. We will revise Eqs.13 and 14 accordingly to express the predictive distribution in terms of $\mathcal{M}$ in the final version. Thank you again for your close reading and for pointing this out.
>
> > * **It is also not clear what the superscript means in Eq. 14, though it can be inferred from context that it indexes Koopman operator samples**
>
> We appreciate the reviewer’s question regarding the superscript notation in Equation 14. To clarify, the superscript on $\mathcal{K}^i$ denotes matrix powers, not sampled Koopman operators — specifically, it refers to recursively applying the Koopman dynamics $i$ times. However, we agree that this notation could be misinterpreted. In response, we have revised the formulation and now present a recursive update rule that more clearly captures multi-step uncertainty propagation:
>
> $$
> \mu_{t+1} = M \mu_t, \quad
> \Sigma_{t+1} = M \Sigma_t M^\top + \Psi \left(1 + \mu_t^\top V \mu_t \right)
> $$
>
> where $\mu_t$ and $\Sigma_t$ are the mean and the covariance at timestep $t$ respectively.
> We will incorporate this clarification and the updated expression in the final version. We thank the reviewer for this helpful observation.
>
> > * **How the context encoder and action encoder parameters are learned, whether it happens within Algorithm 1's loop or via separate pre-training.**
>
> We thank the reviewer for the insightful question. We clarify that both the context and action encoders are trained jointly with the Koopman prior in the same meta-learning loop in Alg. 1 (i.e., no separate pretraining). Gradients from the likelihood objective propagate through the analytic Koopman update into both encoders, enabling them to learn embeddings that support effective single-step Bayesian adaptation. This end-to-end training requires no additional fine-tuning stages or auxiliary losses, resulting in a streamlined and straightforward training pipeline.
>
> > * **The distribution of trajectories is supposedly sampling trajectories under "distinct operating conditions", which is a bit confusing.**
>
> Thank you for raising this point. In our setting—similar to the formulation in [2]—the term “distinct operating conditions” refers to trajectories generated under different latent parameters affecting the system’s dynamics, such as road surface friction, or slopes. Each trajectory corresponds to a short temporal segment from an environment with a specific configuration of these parameters. A detailed discussion of this setup is provided in Lines 117–122 of the Preliminaries section.
>
> In this formulation, we treat each task $\mathcal{T}$ as a random variable indicating which environment a given trajectory segment is drawn from. These variations are not explicitly parameterized, but are implicitly encoded in the trajectory data itself—consistent with common practice in meta-learning.
>
> To evaluate whether our method can also handle tasks drawn from a static environment (i.e., without distribution shift), we included the HalfCheetah-Stationary benchmark in our experiments. In this setting, MetaKoopman maintained strong performance and outperformed adaptive baselines, indicating that the method is robust and not overly reliant on spurious task variation.
>
> We have revised the preliminaries section to clarify the interpretation of $\mathcal{T}$ and better convey this intuition in the corresponding section.
>
> > * **The learning objective for the variational action encoder should be stated more clearly in the main text**
>
> Thank you for highlighting this. The variational action encoder is trained using a combination of the KL regularization term (Eq.17) and the negative log-likelihood (NLL) objective (Eq.15) derived from our Koopman-based trajectory model.
>
> To clarify: the model is trained end-to-end to minimize the following loss:
>
> $$
> \mathcal{L}\_{\text{total}} = \mathcal{L}\_{\text{NLL}} + \mathrm{KL}\left(q\_\phi(\tilde{u} \mid u) || \mathcal{N}(0, I)\right),
> $$
>
> where $\mathcal{L}\_{\text{NLL}}$ corresponds to the negative log-likelihood of the predicted future trajectories under the Koopman dynamics (Eq.15), and the KL term encourages the latent actions $\tilde{u}$ to match a standard Gaussian prior for efficient sampling at test time.
>
> This joint objective ensures that the learned latent space supports both accurate forecasting and scalable planning via the CLAS mechanism. We revised both Section 4.4 and Appendix E to explicitly state this training objective and the interaction between its components.
>
> ---
>
> > ### **W3: As a minor adjustment, above Eq. 15, it should state the meta-objective is to minimise the negative log-likelihood (or maximise the log-likelihood) to match the objective described in the equation.**
>
> We thank the reviewer for raising this point. We would like to note that lines 197–198, right above Eq.~15, mention this: *"The meta-objective is to minimize the log likelihood of the ground truth under the posterior predictive distribution.”* Nevertheless, we will give this clearer spotlight in the paper. We thank the reviewer for the helpful and attentive feedback.
>
> ---
> ---
>
> We now address the reviewer's *questions* below:
>
> > **Q1: How are the parameters of the encoders learned? Are they learned within the same loop in Algorithm 1 or pre-trained via a separate process?**
>
> We thank the reviewer for raising this question. We have addressed this in the 'Weaknesses' section and hope our response clarifies the issue.
>
>
> > **Q2: How is the learning objective for the variational action encoder formulated?**
>
> We appreciate the reviewer for raising this important point. We hope that our answer in the 'Weaknesses' section clarifies the answer.
>
> ---
>
>
> We hope that our responses have addressed your concerns and provided additional clarity on the key points raised. We would be happy to elaborate further if there are any remaining questions.
>
> ---
>
> ##### [1] Singh, Rajpal, Chandan Kumar Sah, and Jishnu Keshavan. "Adaptive Koopman embedding for robust control of complex nonlinear dynamical systems." The International Journal of Robotics Research.
>
> ##### [2] Nagabandi, Anusha, et al. "Learning to Adapt in Dynamic, Real-World Environments through Meta-Reinforcement Learning." International Conference on Learning Representations.

---

> > ### Comment · Reviewer_azPt · 2025-08-04
> >
> > I'd like to thank the authors for addressing most of my concerns.
> > * I'd suggest making the action encoder training objective more precise and clearly stated in the revision. For instance, $u$ in the conditioning $q(\tilde{u} | u)$ is supposedly a single action, but the loss should be minimised over entire trajectories, containing several actions within them. Such revised equation does not make it clear how the loss is evaluated over the entire dataset.
> > * Regarding the text above Eq. 15, it states that "The meta-objective is to *minimize* the log likelihood...". As I was trying to say, one is not supposed to *minimise* the likelihood if their objective is to generate good predictions. You should be *maximising* it, instead. That, however, is equivalent to minimising the *negative* log-likelihood, which is precisely what Eq. 15 describes. The current text, as stated, though, is unfortunately incorrect and potentially misleading, as *log-likelihood* and *negative log-likelihood* are mathematically opposite.

---

> ### Author Response · Authors · 2025-08-05
>
> We thank the reviewer for providing this thoughtful and constructive feedback, and for carefully reviewing our rebuttal. We address the reviewer’s comments below:
>
> ---
>
> > * **I'd suggest making the action encoder training objective more precise and clearly stated in the revision.**
>
> The variational action encoder objective includes a KL divergence term for each future latent action $\tilde{u}_i$, encouraging the distribution of each to match a standard normal prior. Specifically, the complete objective is formulated as:
>
> $$
> \mathcal{L}\_{\text{total}} = \mathcal{L}\_{\text{NLL}} + \sum_{i=t}^{t+H-1} \mathrm{KL}\left(q\_\phi(\tilde{u}_i \mid u_i) || \mathcal{N}(0, I)\right),
> $$
>
> where $H$ denotes the prediction horizon. During inference, the CLAS method samples each latent action $\tilde{u}_i$ directly from the standard normal distribution for trajectory generation. We thank the reviewer for their feedback and will incorporate this revised formulation into the manuscript.
>
> ---
>
> > * **Regarding the text above Eq. 15, it states that "The meta-objective is to minimize the log likelihood..."... The current text, as stated, though, is unfortunately incorrect and potentially misleading, as log-likelihood and negative log-likelihood are mathematically opposite.**
>
> We thank the reviewer for raising this point and apologize for the oversight. Indeed, the objective is to minimize the negative log-likelihood and not the log-likelihood. The updated sentence now reads: *"The meta-objective is to minimize the negative log-likelihood of the ground truth under the posterior predictive distribution."*
>
> Once again, we appreciate the reviewer’s careful reading and constructive feedback.
>
> ---
>
> We sincerely thank the reviewer for the constructive and attentive feedback. We hope that our responses have addressed the concerns raised. Nevertheless, we would be happy to clarify or elaborate on any remaining questions the reviewer may have.

---

> > ### Comment · Reviewer_azPt · 2025-08-05
> >
> > Ok, thanks for the clarification. I guess the loss is evaluated and accummulated over each trajectory during the training process. I'd suggest expanding on the $\mathcal{L}_{\text{NLL}}$ term as well.
> >
> > Given the lack of technical details and citations in Sec. 4.4, I'd also suggest adding references at least on variational autoencoders (VAEs), as it seems that $\mathcal{L}_{\mathrm{total}}$ is simply a variation of the usual evidence lower bound for VAEs, though applying to the specific setting in MetaKoopman for encoding control actions. However, if similar techniques have been used before to model control signals in the control literature or related fields, adding those citations to those references as well should help improving that section by highlighting the soundedness of the approach and connecting it to other methods.

---

> > > ### Author Response · Authors · 2025-08-05
> > >
> > > We thank the reviewer for the constructive feedback. Indeed, the loss is evaluated and accumulated over each trajectory during the training process. To the best of our knowledge, there are no existing techniques that explicitly match action embeddings with a prior and sample latent actions directly from that prior. Nevertheless, we will include citations to the VAE framework and strengthen the connections between our approach and related work in variational inference to better contextualize our method in the revised version.
> > >
> > > Once again, we sincerely thank the reviewer for the thoughtful feedback and thorough discussion of our rebuttal. We hope our responses have meaningfully addressed the concerns raised. We truly appreciate the opportunity to improve the paper through this valuable feedback, and we remain fully open to any further questions or suggestions the reviewer may have.

---

> > > > ### Author Response · Authors · 2025-08-08
> > > >
> > > > **Dear Reviewer,**
> > > >
> > > > This is a kind reminder as we wanted to follow up on our recent discussion regarding our submission, particularly as today is the final day of the rebuttal period.
> > > >
> > > > We sincerely hope that we have addressed your comments and concerns satisfactorily in our last response. Your feedback has been extremely helpful in clarifying and improving the work, and we truly appreciate the time and effort you’ve dedicated to the review process.
> > > >
> > > > If there are any remaining issues or if further clarification is needed, we would be more than happy to provide it.
> > > >
> > > > We also kindly note that, if you find our response satisfactory, an acknowledgment would be greatly appreciated. We hope that the revisions we’ve made help clarify our contributions and may be positively reflected in the evaluation of our submission.

---

### Official Review · Reviewer_WRyM · 2025-07-06

**Clarity:** 3
**Significance:** 3
**Originality:** 2
**Rating:** 4
**Confidence:** 3

**Summary:**

The paper introduces MetaKoopman, a Bayesian meta-learning framework for learning dynamical systems under distribution shifts. It consists of meta-learning a conjugate prior over Koopman operators, and performing closed-form posterior updates from short trajectory segments. Experiments demonstrate that MetaKoopman consistently improves multi-step prediction accuracy, adapts effectively to new dynamics, and enables safer closed-loop control when deployed in the real-world on a heavy-duty autonomous truck.

**Questions:**

See weaknesses.

**Ethical Concerns:**

["NO or VERY MINOR ethics concerns only"]

**Final Justification:**

I lean towards acceptance. The authors adequately addressed my concerns providing new experimental results that strengthen the paper contribution.

**Limitations:**

yes

**Paper Formatting Concerns:**

I have not seen any formatting issue.

**Quality:**

3

**Strengths And Weaknesses:**

Meta-learning is a relevant problem to the community and the presented approach of meta-learning nonlinear dynamics is well-motivated and sound. The paper is clear, well written, and supplemented by extensive experiments including a real-world deployment of the proposed algorithm.

The main weaknesses, in my opinion, are the following:
- From a technical point of view, it is not clear what are the contributions of the paper. Would be good to comment what are the challenges in proving Lemma 4.2 and if there is some additional technical contributions compared to the classical Koopman model results.
- As explicitly stated, MetaKoopman models uncertainty of the Koopman operator but of the input embeddings. I believe this is reasonable in several settings, but could be problematic if embeddings are not trained robustly enough. Thus, I think a more detailed discussions on how the encoder is trained and meta-learned is needed to support the provided results.
- Connected to the point above, it would have been nice to see failure modes of the proposed approach (as well as of other baselines) as a function of the distribution shift, to illustrate the robustness over a spectrum of disturbances.

---

> ### Author Rebuttal · Authors · 2025-07-31
>
> We sincerely thank the reviewer for the thoughtful and constructive feedback. We are encouraged by the reviewer’s positive remarks on the paper’s clear writing, extensive experimental evaluation, and real-world truck deployment, and we appreciate the time you invested in this review. We address the reviewer's concerns below:
>
> ---
>
> > * **W1: From a technical point of view, it is not clear what the specific contributions are, the challenges in proving Lemma 4.2 and what is new compared to classical Koopman results.**
>
> We thank the reviewer for the opportunity to clarify how our approach builds on prior Koopman work and makes new contributions:
>
> 1. **Novel probabilistic formulation.**  Where earlier methods ([1, 2]) learn a single, fixed Koopman operator, we place a distribution over all plausible operators and meta-learn this distribution. Online, this distribution is updated via closed-form Bayesian inference, enabling rapid adaptation to distribution shifts (e.g., our truck switching between asphalt and ice) while explicitly capturing uncertainty in the system dynamics through uncertainty in the operator’s parameters—capabilities that classical Koopman models lack.
>
> 2. **Analytic propagation of that uncertainty.**  Lemma 4.2 shows how this parameter uncertainty maps—in closed form—into a posterior predictive distribution over future states, providing principled uncertainty quantification in the predictions.
>
> 3. **Real-time planning & first low-friction deployment.**  A variational action encoder lets the planner sample actions directly in latent space, dramatically reducing rollout time. To the best of our knowledge, this is also the first time these models are demonstrated on low-friction autonomous driving.
>
> A detailed list of our contributions is presented in lines 62–77 of the paper, which we hope addresses the reviewer’s concern.
>
> ---
>
> > * **W2: MetaKoopman captures uncertainty in the Koopman operator rather than input embeddings. Therefore, a more detailed explanation of the encoder’s training and meta-learning process is necessary to validate the results.**
>
> We thank the reviewer for the insightful question. We clarify that both the context and action encoders are trained jointly with the Koopman prior in the same meta-learning loop in Alg. 1 (i.e., no separate pretraining).
>
> Below, we outline how these components interact during training and explain why this design leads to strong predictive performance and well-calibrated uncertainty.
>
> **How the pieces are trained—one loop, three parameter blocks**
>
> 1. **Embed**
>    - A context encoder $G_{\theta_\text{ctx}}$ turns a $q$-step history trajectory $(x_{t-q:t}, u_{t-q:t})$ into latent matrix $\mathcal{Z}$.
>    - An action encoder $H_{\theta_\text{act}}$ maps planned future controls $u_{t+1:t+h}$ into latent action embeddings.
>
> 2. **Adapt**
>    - The history embeddings $\mathcal{Z}$ drive the *closed-form* Bayesian update (Eq.~10) that adapts the Koopman prior  $\theta_k = \\{M, V, \nu, \Psi, \beta\\} $ into task-specific parameters:  $ \theta_k'(\mathcal{D}^{\text{tr}}_{\mathcal{T}}) = \text{Adapt}(\theta_k,\mathcal{Z}). $
>
> 3. **Predict**
>    - The adapted operator $\mathcal{K}(\theta'_k)$ and the future-action embeddings are combined using Lemma 2 to forecast future state predictions.
>
> 4. **Optimise**
>    - The **only** objective we use is the task-level negative log-likelihood in Eq.~15. Gradients from this objective flow through the analytic Koopman update and into both encoders, so $G$ and $H$ are trained to produce embeddings that enable a single-step Bayesian adaptation to generalise well. No extra fine-tuning stages or auxiliary losses are used.
>
> **Empirically,** MetaKoopman achieves strong predictive performance across a range of distribution shifts (Tables 1 and 2), without requiring test-time encoder updates.
>
> **The model's uncertainty accounts for suboptimal embeddings by design.** We use the posterior predictive variance to measure when the Koopman operator has difficulty in adapting to the past trajectory segment. This behavior is consistent with our observed correlation between predictive uncertainty and error (Table 3), indicating that the model expresses higher uncertainty when adaptation is less reliable.
>
> We will highlight this training structure and its implications for robustness more clearly in the future version of the manuscript.
>
> ---
>
> > * **W3:  It would have been nice to see failure modes of the proposed approach (as well as of other baselines) as a function of the distribution shift, to illustrate the robustness over a spectrum of disturbances.**
>
> Thank you for this excellent suggestion to evaluate robustness under increasing distribution shift. In response, we conducted a targeted experiment in the **Hopper** environment to assess how each method handles significant deviations in gravity.
>
> Models were trained on gravity values sampled within a narrow range of $\pm 1.5$ around the nominal setting. For evaluation, we tested each method under a series of increasing gravity shifts: $[0, 1, 2, 3, 4]$ added to the default gravity, simulating progressively more challenging out-of-distribution dynamics.
>
> To enable fair comparison, the multi-step prediction MSEs are **normalized within each column**: values are reported relative to the method’s own performance at shift = 0. **Note:** The first-row values are all `1.0` by construction, but the *underlying absolute MSE differs across methods*.
>
> | Gravity Shift | MetaKoopman | BMAML     | GrBAL     | DKO       | BNN       | EMLP      | NODE      |
> |---------------|-------------|-----------|-----------|-----------|-----------|-----------|-----------|
> | +0            | **1**       | **1**     | **1**     | **1**     | **1**     | **1**     | **1**     |
> | +1            | 1.021       | 1.085     | **1.017** | 1.195     | 1.082     | 1.195     | 1.099     |
> | +2            | **1.455**   | 1.534     | 1.555     | 2.080     | 2.122     | 2.080     | 1.388     |
> | +3            | **1.866**   | 1.918     | 2.645     | 4.938     | 2.419     | 4.938     | 2.061     |
> | +4            | **2.368**   | 2.903     | 2.850     | 4.538     | 3.586     | 4.538     | 3.661     |
>
> As shown, **MetaKoopman** demonstrates the most gradual increase in error across all shift levels, outperforming all baselines in most rows. This highlights its ability to generalize under physical perturbations, benefiting from its closed-form Bayesian adaptation and meta-learned structure. We thank the reviewer so much for the suggestion and will include this additional evaluation in the updated manuscript.
>
> ---
>
> We sincerely thank the reviewer for the thoughtful and constructive feedback. We hope our responses have adequately addressed the main concerns and clarified our contributions. We are happy to provide any further clarification or respond to any additional questions the reviewer may have.
>
> ---
>
> ##### [1] Shi, Haojie, and Max Q-H. Meng. "Deep Koopman operator with control for nonlinear systems." IEEE Robotics and Automation Letters 7.3 (2022): 7700-7707.
>
> ##### [2] Takeishi, Naoya, Yoshinobu Kawahara, and Takehisa Yairi. "Learning Koopman invariant subspaces for dynamic mode decomposition." Advances in neural information processing systems 30 (2017).

---

### Note · Authors · 2025-08-16

We thank the reviewers (R1 → WRyM, R2 → azPt, R3 → wjtk, R4 → cqV4) and the AC for their thoughtful feedback and for taking the time to engage with our rebuttal. We hope our clarifications and additional analyses addressed the concerns and highlighted the strengths of our work.

---

### Clarifications
We made the following clarifications:
- Clarified that Eq. 15 **minimizes negative log-likelihood** (R2).
- Clarified that the predictive distribution (``Eqs. 13–14``) depends on the mean of the Koopman distribution, with recursive mean/variance added for multi-step prediction (R2).
- Emphasized that context and action encoders are **trained jointly** in ``Algorithm 1`` (R1, R2).
- Expanded the **variational action encoder objective**, stating full NLL + KL in the main text (R2).
- Broadened related work to connect with adaptive Koopman methods (Singh et al., 2025), robust control, and test-time adaptation (R2, R4).
- Highlighted **ablations on tempering and history length** (``Tables 4–5``) (R4).

---

### Additional Experiments
During the rebuttal, we:
- Added **Hopper–Gravity shift-spectrum** evaluation, showing MetaKoopman degrades most gracefully under stronger shifts (R1).
- Added **7-DoF robotic manipulation (Panda-damping)**, extending to high-dimensional manipulation (R3, R4).
- Evaluation now spans **7 environments, 3 modalities** (driving, locomotion, manipulation), broadening coverage (R3, R4).

---

### Contributions of the Paper
- First to introduce **closed-form Bayesian meta-learning of Koopman operators**, deriving analytic posterior and predictive for calibrated multi-step forecasting.
- Unifies **Koopman (structure), closed-form Bayes (uncertainty + efficiency), and meta-learning (adaptation)** in one framework, enabling real-time planning via the variational action encoder and the CLAS mechanism.
- Demonstrated real-world safety impact in **truck trials** on snow, ice, and μ-split, where online adaptation and calibrated uncertainty improved safety (R3, R4).
- Benchmarked against **six baselines** spanning meta-learning, Bayesian meta-learning, Koopman, continuous-time, and uncertainty modeling, with consistent gains in accuracy (``Tables 1–2``), uncertainty (``Table 3``), and runtime (``Table 6``) (R4).

- Ablated adaptation (``Fig. 2``), history length (``Table 5``), and tempering (``Table 4``) (R4).

---

We again thank the AC and reviewers for constructive feedback, which strengthened the clarity, scope, and positioning of the paper.

---

### Decision · Program_Chairs · 2025-09-17

**Decision:**

Accept (poster)

**Comment:**

This paper proposes meta learning of a koopman operator for control of
a nonlinear under domain shift.

The strengths of the paper include proposing a novel method, testing
it on a real semi truck, and doing an extensive simulation comparison
to other methods.